# Protein kinase Cδ is essential for the IgG response against T-cell-independent type 2 antigens and commensal bacteria

Saori Fukao, Kei Haniuda, Hiromasa Tamaki, Daisuke Kitamura*

Division of Cancer Cell Biology, Research Institute for Biomedical Sciences (RIBS), Tokyo University of Science, Noda, Japan

**Abstract** Antigens (Ags) with multivalent and repetitive structure elicit IgG production in a T-cell-independent manner. However, the mechanisms by which such T-cell-independent type-2 (TI-2) Ags induce IgG responses remain obscure. Here, we report that B-cell receptor (BCR) engagement with a TI-2 Ag but not with a T-cell-dependent (TD) Ag was able to induce the transcription of *Aicda* encoding activation-induced cytidine deaminase (AID) and efficient class switching to IgG3 upon costimulation with IL-1 or IFN-α in mouse B cells. TI-2 Ags strongly induced the phosphorylation of protein kinase C (PKC)δ and PKCδ mediated the *Aicda* transcription through the induction of BATF, the key transcriptional regulator of *Aicda*. In PKCδ-deficient mice, production of IgG was intact against TD Ag but abrogated against typical TI-2 Ags as well as commensal bacteria, and experimental disruption of the gut epithelial barrier resulted in fatal bacteremia. Thus, our results have revealed novel molecular requirements for class switching in the TI-2 response and highlighted its importance in homeostatic commensal-specific IgG production.

## Editor's evaluation

Although how structurally different antigens can activate B-lymphocytes to produce IgG with or without T cell help has been a long-standing question, this manuscript provides a clue to this important issue.

*For correspondence:
kitamura@rs.tus.ac.jp

Competing interest: The authors declare that no competing interests exist.

## Introduction

Ag-specific antibody production is essential for humoral immunity. After T-cell-dependent (TD) antigen (Ag) exposure, B cells are activated by interacting with cognate T cells and then proliferate, undergo class switching, and differentiate into plasma cells (PCs) and memory cells. In contrast, T-cell-independent (TI) Ags activate B cells without cognate help of T cells. TI type 1 (TI-1) Ags engage Toll-like receptors (TLRs) in addition to the B-cell receptor (BCR) whereas TI-2 Ags extensively crosslink the BCR because of their highly repetitive structure (*Mond et al., 1995*). In addition, some help from non-T cells, such as dendritic cells and innate lymphoid cells, may support B cells in a TI response (*Magri et al., 2014*; *Balázs et al., 2002*). Following activation, TI Ags induce proliferation, class switching, and antibody production by B cells. TI-2 Ags are large multivalent molecules, such as bacterial capsular polysaccharides and viral capsids, and thus antibody production against polysaccharides by the TI-2 response confers protection against disease (*Mond et al., 1995*; *Lesinski and Westerink, 2001*). To date, however, the mechanism of B-cell activation in the TI-2 response is poorly understood as compared to that in TD and TI-1 responses.

Engagement of the BCR by Ag activates various signaling cascades and promotes B-cell survival, proliferation and differentiation. The antibody production in a TI-2 response, but not in a TD response,

**eLife digest** When the human body faces a potentially harmful microorganism, the immune system responds by finding and destroying the pathogen. This involves the coordination of several different parts of the immune system. B cells are a type of white blood cell that is responsible for producing antibodies: large proteins that bind to specific targets such as pathogens. B cells often need help from other immune cells known as T cells to complete antibody production.

However, T cells are not required for B cells to produce antibodies against some bacteria. For example, when certain pathogenic bacteria coated with a carbohydrate called a capsule – such as pneumococcus, which causes pneumonia, or salmonella – invade our body, B cells recognize a repetitive structure of the capsule using a B-cell antigen receptor. This recognition allows B cells to produce antibodies independently of T cells. It is unclear how B cells produce antibodies in this situation or what proteins are required for this activity.

To understand this process, Fukao et al. used genetically modified mice and their B cells to study how they produce antibodies independently of T cells. They found that a protein called PKCδ is critical for B cells to produce antibodies, especially of an executive type called IgG, in the T-cell-independent response. PKCδ became active when B cells were stimulated with the repetitive antigen present on the surface of bacteria like salmonella or pneumococcus. Mice that lack PKCδ were unable to produce IgG independently of T cells, leading to fatal infections when bacteria reached the tissues and blood.

Understanding the mechanism behind the T cell-independent B cell response could lead to more effective antibody production, potentially paving the way for new vaccines to prevent fatal diseases caused by pathogenic bacteria.

depends upon proximal BCR signaling molecules such as Btk and BLNK (*Khan et al., 1995*; *Xu et al., 2000*; *Fruman et al., 2000*), suggesting that BCR signaling plays a more critical role for B-cell activation in a TI-2 response than in a TD response. Given the highly repetitive structure of TI-2 Ags and the high demands for BCR signaling in TI-2 responses, TI-2 Ags seem to induce BCR signaling and the subsequent response more strongly than TD Ags. However, such functional differences between TD Ags and TI-2 Ags have not been investigated.

Class (isotype) switching of the immunoglobulin (Ig) on B cells from IgM to either IgG, IgE, or IgA is caused by selective recombination of Igh constant region ($C_H$) genes, namely class-switch recombination (CSR). Selection of the $C_H$ gene to be recombined is determined by T-cell-derived cytokines in the TD response but the mechanism in the TI-2 response is not clear. Immunization with TI-2 Ags, including NP-Ficoll and bacterial polysaccharides, predominantly elicits IgG3 (*Perlmutter et al., 1978*; *Slack et al., 1980*; *Rubinstein and Stein, 1988*) and IgG3 is required for protection against pneumococcal infection (*McLay et al., 2002*). CSR absolutely requires activation-induced cytidine deaminase (AID) (*Muramatsu et al., 2000*). AID deaminates deoxycytidine and introduces DNA double strand breaks (DSBs) in the switch (S) regions lying 5′ of the Cμ gene and the other targeted $C_H$ gene by triggering the DNA repair machinery (*Stavnezer et al., 2008*). CSR proceeds through looping-out deletion of the DNA segment intervening between Sμ and the other S region from the chromosome and religation of the DSB-free ends in the two S regions. Consequently, it leads to replacement of the Cμ gene with a different $C_H$ gene downstream of the variable region exon in the *Igh* locus.

In a TD immune response, stimulation with CD40L and cytokines, such as IL-4 and TGFβ, induce the expression of AID and subsequent CSR in the B cell (*Vaidyanathan et al., 2014*; *Dedeoglu et al., 2004*), while signaling through TLRs, BCR, and TACI induce the expression of AID and CSR in a TI-1 response (*Xu et al., 2012*). On the other hand, the signaling and molecular demands for the induction of AID and CSR in the TI-2 response are far less understood, presumably due to the lack of in vitro studies that mimic a TI-2 response. Although TACI is required for IgG production in a TI-2 response (*von Bülow et al., 2001*), stimulation of TACI alone induces the expression of AID and IgG production only modestly (*Castigli et al., 2005*). Therefore, signaling through other receptors besides TACI seems to be required. As antibody production itself is disrupted in mice lacking proximal BCR signaling molecules (*Khan et al., 1995*; *Xu et al., 2000*), the requirement of BCR signaling and the downstream molecules for CSR remain unclear. Here, we report our finding that stimulation of the BCR with NP-Ficoll, a typical TI-2 Ag, triggered IgG CSR in the presence of secondary stimulation

by IL-1, IFNα, or TLR ligands in vitro and that protein kinase C (PKC)δ is critical for the Ag-mediated upregulation of AID and IgG production in the TI-2 response.

## Results

### A TI-2 Ag induces B-cell proliferation and potentiates CSR to IgG

Considering the unique structure of TI-2 Ags, it is plausible that the engagement of the BCR with TI-2 Ags and TD Ags differently induces downstream signaling that leads to B-cell activation, although this idea has not been tested properly so far. We tested this in vitro by stimulating NP-specific B cells with the TD Ag NP-CGG or the TI-2 Ag NP-Ficoll to compare their ability to induce signaling, proliferation, and antibody production. NP-specific B cells were prepared from $Igk^{-/-}$ mice (expressing only $\lambda$ isotype light chain) carrying a $V_H$ B1-8 knock-in gene encoding a $V_H$ region which binds to NP when coupled with a $\lambda$ light chain. Although NP-CGG induced little proliferation and no IgM production, NP-Ficoll induced strong proliferation and IgM production, whereas neither induced IgG production (*Figure 1A and B*). Thus, although NP-Ficoll alone can strongly activate B cells, additional stimulation seemed to be required for induction of class switching, similar to a previous report that anti-δ mAb/dextran and TLR ligands synergistically induced AID and CSR (*Pone et al., 2012*). Indeed, NP-Ficoll induced IgG production, generation of IgG+ cells and *Aicda* transcription in the presence of TLR ligands, LPS, R-848, or CpG, while NP-CGG did so marginally (*Figure 1—figure supplement 1A-C*). Besides TI-1 Ags, such as TLR ligands, we sought to identify costimulating molecules that induce class switching in the TI-2 response, some of which were suggested previously (*Magri et al., 2014*; *Balázs et al., 2002*). Since IgG3 is the most dominant class-switched Ig isotype produced in a TI-2 response, we screened various cytokines for their ability to promote the production of IgG3 in the presence of NP-Ficoll, and identified IL-1α, IL-1β, and IFNα as efficient costimuli for IgG3 production (*Figure 1—figure supplement 1D* and *Figure 1C*). These cytokines, together with NP-Ficoll, induced generation of IgG3+ cells (*Figure 1D*), as well as IgG1+ and IgG2b+ cells to a lesser extent (*Figure 1—figure supplement 1E*). In the presence of these cytokines, NP-Ficoll was far more potent than NP-CGG for IgG3 production, IgG3+ cell generation, Sμ-Sγ3 CSR (detectable as the Iγ3-Cμ transcript from the switch circle DNA) (*Xu et al., 2012*; *Kinoshita et al., 2001*), and the induction of *Aicda* transcription (*Figure 1C–E*). NP-Ficoll or each of these cytokines alone could not induce those responses (*Figure 1C–E*). Collectively, these results indicate that BCR signaling elicited by TI-2 Ag engagement is pivotal for the TI-2 B-cell response, namely induction of proliferation and antibody production, as well as potentiation of CSR to IgG.

### PKCδ is required for IgG production and AID expression induced by TI-2 Ag stimulation

By Western blot analyses for BCR signaling using NP-specific B cells, we noticed that NP-Ficoll-induced phosphorylation of PKCδ tyrosine 311 (Y311), indicative of activation of the kinase (*Balasubramanian et al., 2006*), more strongly than NP-CGG (*Figure 2A*). PKCδ belongs to a novel PKC subfamily (*Salzer et al., 2016*), whose function in the immune response remains elusive. Thus, we analyzed the role of PKCδ in the TI-2 response using PKCδ-deficient (*Prkcd*$^{-/-}$) mice. B cells from *Prkcd*$^{-/-}$ mice normally proliferate and produce IgM when stimulated with NP-Ficoll alone (*Figure 2—figure supplement 1A, B*). Upon costimulation with NP-Ficoll and IL-1α, IL-1β, and IFNα, *Prkcd*$^{-/-}$ B cells produced comparable levels of IgM but markedly reduced levels of IgG3 compared to *Prkcd*$^{+/+}$ cells (*Figure 2B*). As class switching occurs along with cell division (*Deenick et al., 1999*), we analyzed the frequency of IgG3+ cells at each cell division using B cells labeled with CellTrace Violet (CTV). While cell division was almost equivalent between these cells after costimulation with any of the cytokines and NP-Ficoll (*Figure 2—figure supplement 1C*), the frequencies of IgG3+ cells barely increased at any points of cell division in *Prkcd*$^{-/-}$ cells in contrast to *Prkcd*$^{+/+}$ cells (*Figure 2C*). Therefore, PKCδ is necessary for generation of IgG3+ cells regardless of cell division. Accordingly, *Igh* locus CSR to IgG3 assessed by the circle Iγ3-Cμ transcript and the expression of *Aicda* transcripts were attenuated in *Prkcd*$^{-/-}$ B cells (*Figure 2D*). Collectively, these results suggest that PKCδ mediates the expression of AID and class switching to IgG3 induced by BCR stimulation with TI-2 Ag and IL-1/IFNα costimulation.

*Prkcd*$^{-/-}$ B cells also exhibited defects in the production of IgG and *Aicda* transcripts, but normal IgM production, upon stimulation with NP-Ficoll and TLR ligands (*Figure 2—figure supplement 1D,*

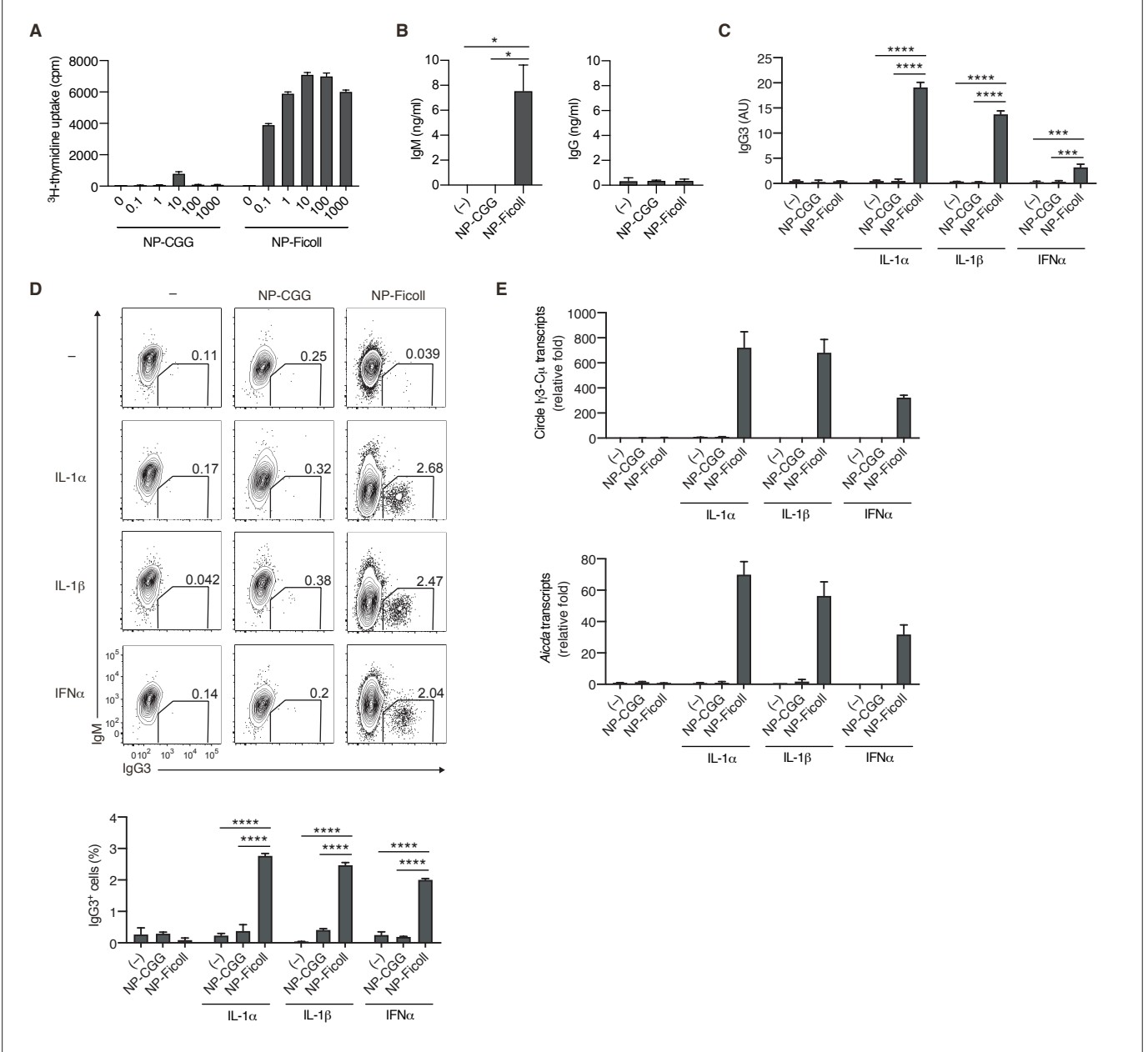

**Figure 1.** T-cell-independent type-2 (TI-2) antigen (Ag) distinctively induces B-cell activation and class-switch recombination (CSR) to IgG in vitro. (**A and B**) Naive B cells purified from the spleen of *Igk⁻/⁻ B1-8ᶠˡᵒˣ/⁺* mice were stimulated with NP-CGG or NP-Ficoll. (**A**) ³H-thymidine incorporation on day 3. (**B**) IgM and IgG concentrations in the culture supernatants on day 3. (**C–E**) *Igk⁻/⁻ B1-8ᶠˡᵒˣ/⁺* B cells were stimulated with none (–), NP-CGG, or NP-Ficoll and IL-1α, IL-1β, or IFNα, as indicated. (**C**) Enzyme-linked immunosorbent assay (ELISA) of IgG3 in the culture supernatant on day 3. AU, arbitrary units. (**D**) Representative flow cytometric plots on day 3 with the numbers indicating percentages of IgG3⁺ B cells (top). The frequencies of IgG3⁺ cells among the B cells (bottom). (**E**) quantitative reverse-transcription polymerase chain reaction (qRT-PCR) analysis of the circle I γ 3-Cμ and *Aicda* transcripts on day 2. Data are means ± standard deviations (SDs) of two (**B and C**), two to three (**D**), or three (**A**) biological replicates or three technical replicates (**E**). The data are representative of at least three (**A and B**) or two (**C–E**) independent experiments. *p < 0.05; ***p < 0.001; ****p < 0.0001; p values were calculated by one- (**B**) or two (**C and D**)-way analysis of variance (ANOVA) with Tukey's test.

The online version of this article includes the following figure supplement(s) for figure 1:

**Source data 1.** Source data for *Figure 1A–E*.

**Figure supplement 1.** T-cell-independent type-2 (TI-2) antigen (Ag) induces the generation of IgG⁺ cells in the presence of secondary stimulation.

**Figure supplement 1—source data 1.** Source data for *Figure 1—figure supplement 1A, C, D*.

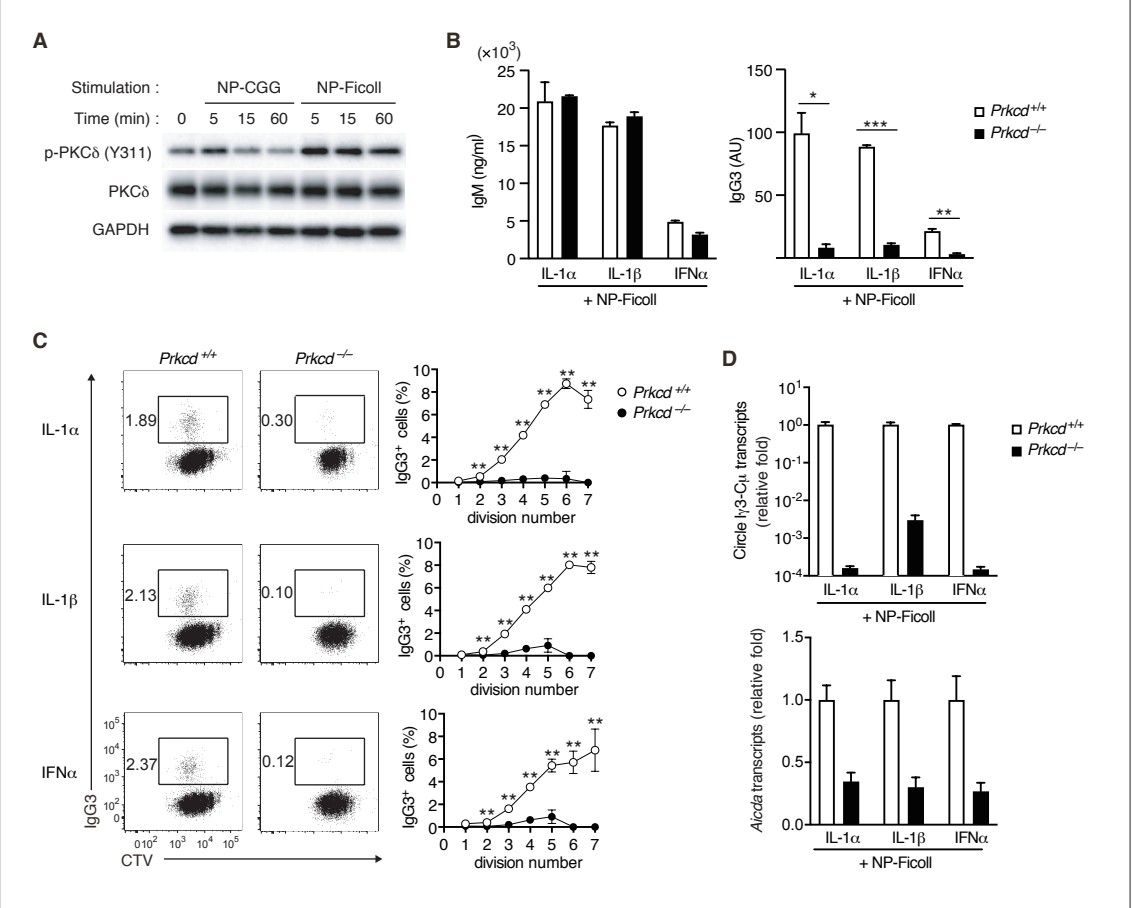

**Figure 2.** Impaired T-cell-independent type-2 (TI-2) antigen (Ag)-mediated class-switch recombination (CSR) in protein kinase C (PKC) δ -deficient B cells. (**A**) Immunoblot analysis of the indicated molecules in *Igk⁻/⁻ B1-8^{flox/+}* B cells stimulated with NP-CGG or NP-Ficoll for the indicated time periods. (**B–D**) *Prkcd^{+/+} Igk⁻/⁻ B1-8^{flox/+}* or *Prkcd⁻/⁻ Igk⁻/⁻ B1-8^{flox/+}* B cells were stimulated with NP-Ficoll and IL-1α, IL-1β, or IFNα. The B cells were labeled with CellTrace Violet (CTV) before culture in C. (**B**) Enzyme-linked immunosorbent assay (ELISA) of IgM and IgG3 in the culture supernatants on day 5. AU, arbitrary units. (**C**) Representative flow cytometric plots of the B cells on day 3 showing percentages of IgG3⁺ B cells (left). The frequencies of IgG3⁺ cells at each cell division number (right). (**D**) qRT-PCR analysis of the circle I γ 3-Cμ and *Aicda* transcripts on day 2. Data are means ± standard deviations (SDs) of two (**B**) or three (**C**) biological replicates or three technical replicates (**D**). The data are representative of at least three (**A and B**) or two (**C and D**) independent experiments. *p < 0.05; **p < 0.01; ***p < 0.001; p values were calculated by unpaired multiple *t*-test (**B and C**).

The online version of this article includes the following figure supplement(s) for figure 2:

**Source data 1.** Source data for *Figure 2B–D*.

**Source data 2.** Source data for *Figure 2A*.

**Source data 3.** Source data for *Figure 2A*.

**Source data 4.** Source data for *Figure 2A*.

**Source data 5.** Source data for *Figure 2A*.

**Figure supplement 1.** In vitro response of protein kinase C (PKC) δ -deficient B cells.

**Figure supplement 1—source data 1.** Source data for *Figure 2—figure supplement 1A, B, D-G*.

**E**), but no defects in the production of IgM, IgG, and *Aicda* transcripts, upon stimulation with TLR ligands alone (*Figure 2—figure supplement 1F, G*). Together with the above results, PKCδ appears to be selectively required for TI-2-Ag-mediated BCR signaling to potentiate CSR to produce IgG.

## PKCδ mediates IgG production in a TI-2 response in vivo

We next evaluated the contribution of PKCδ to IgG production in a TI-2 immune response using mice carrying loxP-flanked *Prkcd* alleles and *Cd19*-cre allele that lack *Prkcd* specifically in B cells (referred to herein as *Cd19^{cre/+} Prkcdf^{f/f}*). We first examined the cellularity of mature B-cell subpopulations in

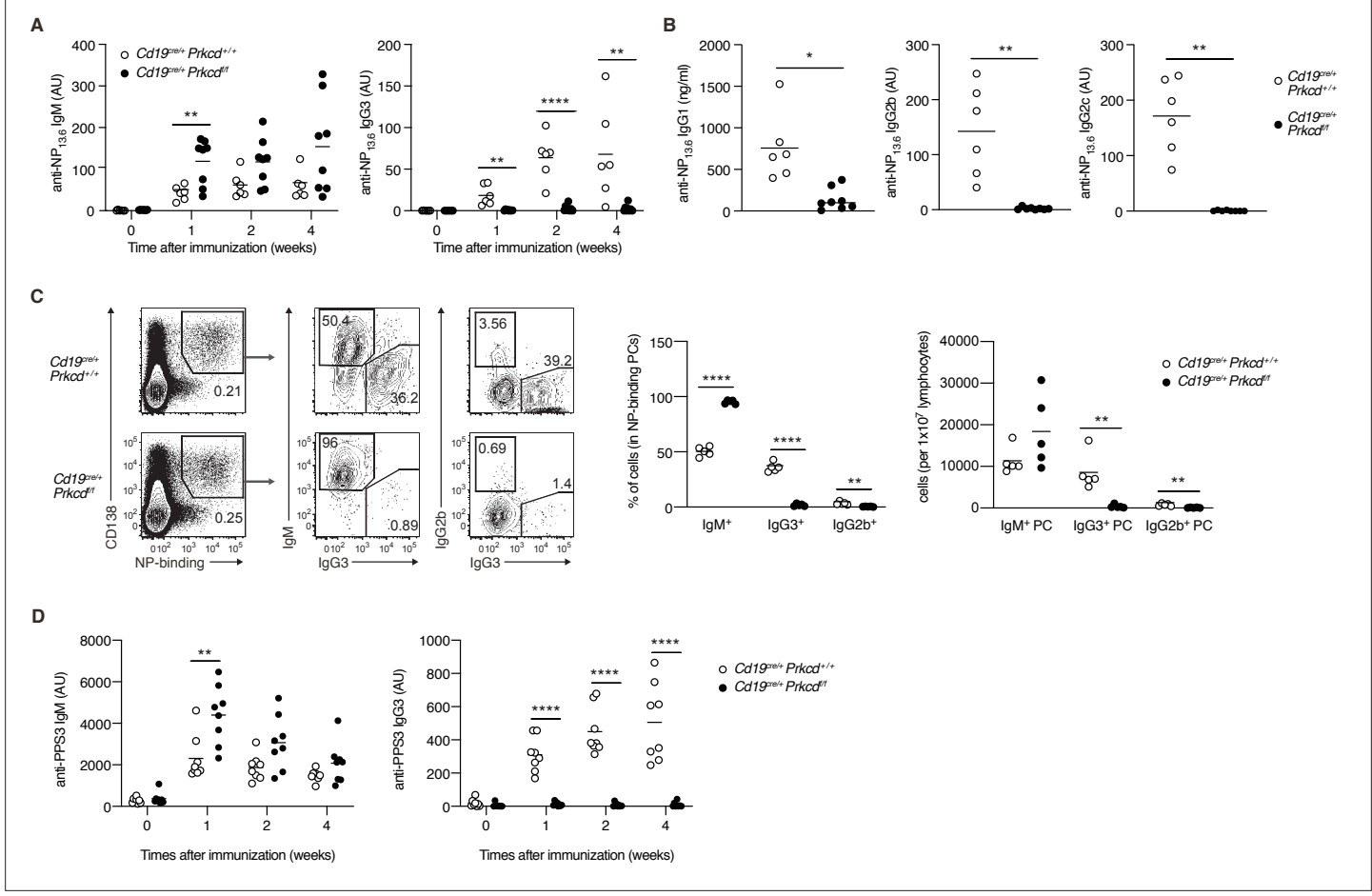

**Figure 3.** Protein kinase C (PKC) δ is required for IgG production in T-cell-independent type-2 (TI-2) response in vivo. (**A–C**) $Cd19^{cre/+} Prkcd^{+/+}$ and $Cd19^{cre/+} Prkcd^{f/f}$ mice were immunized with NP-Ficoll. (**A**) Enzyme-linked immunosorbent assay (ELISA) of serum anti-NP IgM and IgG3 at the indicated weeks. (**B**) ELISA of serum anti-NP IgG1, IgG2b, and IgG2c at 2 weeks after immunization. (**C**) Representative flow cytometric plots of the spleen cells on day 7 after immunization. The numbers indicate the percentage of cells in each gate (left). The frequencies of IgM+, IgG2b+, and IgG3+ cells among NP-binding plasma cells (PCs; NP+ CD138+; middle) and the numbers of such cells per $1 \times 10^7$ total lymphocytes (right) are plotted. (**D**) ELISA of anti-PPS3 IgM and IgG3 in the serum of $Cd19^{cre/+} Prkcd^{+/+}$ and $Cd19^{cre/+} Prkcd^{f/f}$ mice at the indicated weeks after immunization with PPS3. Results are presented in AU, arbitrary units (**A, B, and D**). Small horizontal bars are the means of six to eight (A and B), five (**C**), and eight (**D**) biological replicates. Each symbol represents an individual mouse. The data are representative of three (**A and C**) or two (**B and D**) independent experiments. *p < 0.05; **p < 0.01; ****p < 0.0001; p values were calculated by unpaired multiple (**A, C, and D**) or two-tailed unpaired Welch's t-test (**B**).

The online version of this article includes the following figure supplement(s) for figure 3:

**Source data 1.** Source data for *Figure 3A–D*.

**Figure supplement 1.** B-cell development and T-cell-dependent (TD) response of protein kinase C (PKC) δ -deficient mice.

**Figure supplement 1—source data 1.** Source data for *Figure 3—figure supplement 1A, B*.

such mice and their control ($Cd19^{cre/+} Prkcd^{+/+}$). The numbers of follicular and marginal zone B cells in spleens were comparable between these mice, whereas those of splenic and peritoneal B1 cell populations were slightly increased in $Cd19^{cre/+} Prkcd^{f/f}$ mice (*Figure 3—figure supplement 1A*). Thus, PKCδ is dispensable for the development of B cells, as previously described (*Mecklenbräuker et al., 2002*; *Miyamoto et al., 2002*).

To analyze the role of PKCδ in a TI-2 response, we immunized $Cd19^{cre/+} Prkcdf^{f/f}$ and the control mice with NP-Ficoll. Although the production of serum anti-NP IgM was slightly enhanced in $Cd19^{cre/+} Prkcdf^{f/f}$ mice, that of anti-NP IgG3 was severely suppressed (*Figure 3A*), and so was that of anti-NP IgG1, IgG2b, and IgG2c (*Figure 3B*). We then analyzed PCs in the spleen 1 week after immunization. Among the NP-specific PCs, the proportion and the number of IgG3- and IgG2b-producing PCs were severely decreased in $Cd19^{cre/+} Prkcdf^{f/f}$ mice compared to the control mice, whereas the number of

IgM$^+$ PCs was comparable between the two groups (*Figure 3C*). Therefore, PKCδ is required for the generation of IgG$^+$ PCs and the subsequent production of IgG in the TI-2 response.

The capsular polysaccharides of *Streptococcus pneumoniae*, such as pneumococcal polysaccharide serotype 3 (PPS3), are also classified as TI-2 Ags, and immunization with PPS3 is known to induce an Ag-specific IgG3 response (*McLay et al., 2002*). Thus, we next immunized *Cd19$^{cre/+}$ Prkcdf$^{f/f}$* and control mice with PPS3. Although early production of anti-PPS3 IgM was modestly enhanced in *Cd19$^{cre/+}$ Prkcdf$^{f/f}$* mice, production of anti-PPS3 IgG3 was ablated in these mice (*Figure 3D*). Thus, PKCδ appears to be generally required for IgG production in response to a variety of TI-2 Ags.

We next assessed whether PKCδ is also required for IgG production in a TD response. After immunization with NP-CGG, the production of anti-NP IgM was transiently enhanced in *Cd19$^{cre/+}$ Prkcdf$^{f/f}$* mice at 1 week, whereas anti-NP IgG1 and IgG3 titers were normal in these mice (*Figure 3—figure supplement 1B*). These data indicated that PKCδ signaling is required for IgG production in a TI-2 response, but not in a TD response.

## PKCδ mediates class switching through induction of AID in a TI-2 response in vivo

We next assessed whether PKCδ mediates IgG production through class switching in an in vivo TI-2 response. To discriminate Ag-specific B-cell responses, we transferred CTV-labeled naive B cells from *B1-8$^{hi}$* CD45.1 mice into C57BL/6 (B6) mice, which were immunized with NP-Ficoll on the next day and analyzed by flow cytometry 3 days later. About 30 % of the splenic donor cells were IgG3$^+$ in the mice transferred with control B cells, whereas only about 3 % were IgG3$^+$ in recipients of *Cd19$^{cre/+}$ Prkcdf$^{f/f}$* B cells (*Figure 4A*). IgG3$^+$ cells emerged at the second cell division and their frequency was increased as the cell division proceeded in the cells of control mice, whereas the frequency of IgG3$^+$ cells was extremely low in the cells of *Cd19$^{cre/+}$ Prkcdf$^{f/f}$* mice at any point of cell division (*Figure 4A*). Thus, PKCδ mediates the generation of IgG3$^+$ cells in a manner unrelated to cell division.

We next analyzed molecular events associated with CSR in sorted donor cells (*Figure 4—figure supplement 1A*). The amounts of the Iγ3-Cμ circle transcripts and the Iμ-Cγ3 early postswitch transcripts were far less in *Cd19$^{cre/+}$ Prkcdf$^{f/f}$* B cells compared to control cells, indicating fewer CSR events in the former in vivo (*Figure 4B*). The expression of Iμ-Cμ and Iγ3-Cγ3 germline transcripts was comparable between the *Cd19$^{cre/+}$ Prkcdf$^{f/f}$* B cells and the control cells, whereas the expression of *Aicda* was substantially reduced in the former (*Figure 4B*). These results indicate that PKCδ mediates CSR to IgG3 through upregulation of AID mRNA, but not through the activation of *Ig* gene S regions in B cells responding to TI-2 Ags in vivo.

To examine whether the reduction of AID is a primary reason for the impaired generation of IgG3$^+$ cells from PKCδ-deficient B cells, we transduced AID into in vivo primed *Cd19$^{cre/+}$ Prkcdf$^{f/f}$ B1-8$^{hi}$* B cells and transferred them into B6 mice that had been immunized with NP-Ficoll 1 day previously (*Figure 4—figure supplement 1B*). The reconstitution of AID expression restored IgG3 class switching to a frequency comparable to the same cells reconstituted with PKCδ (*Figure 4—figure supplement 1C* and *Figure 4C*). Collectively, these results indicate that PKCδ mediates class switching to IgG3 by upregulating the expression of AID in the TI-2 response.

## PKCδ upregulates the transcription of *Aicda* through BATF

It has been shown that expression of *Aicda* gene is regulated by various transcriptional factors (*Vaidyanathan et al., 2014*; *Tran et al., 2010*; *Crouch et al., 2007*). To assess the role of PKCδ in *Aicda* gene expression, we quantified expression levels of the genes encoding such transcription factors in PKCδ-sufficient and PKCδ-deficient B cells collected from mice immunized with NP-Ficoll. Among the genes tested, we found that the amount of *Batf* mRNA was markedly lower in *Cd19$^{cre/+}$ Prkcdf$^{f/f}$* B cells than in control cells (*Figure 5A*). The expression of BATF mRNA and protein was induced by in vitro stimulation with NP-Ficoll alone in PKCδ-sufficient B cells, but only marginally in PKCδ-deficient B cells (*Figure 5B and C*), whereas BATF expression was not induced by NP-CGG (*Figure 5B* and *Figure 5—figure supplement 1A*). IL-1α, IL-1β, or IFNα did not augment the phosphorylation of PKCδ nor induce *Batf* expression nor enhance the NP-Ficoll-induced *Batf* expression (*Figure 5B* and *Figure 5—figure supplement 1B-D*). Collectively, these data indicate that PKCδ induces the expression of BATF downstream of the BCR in the TI-2 response.

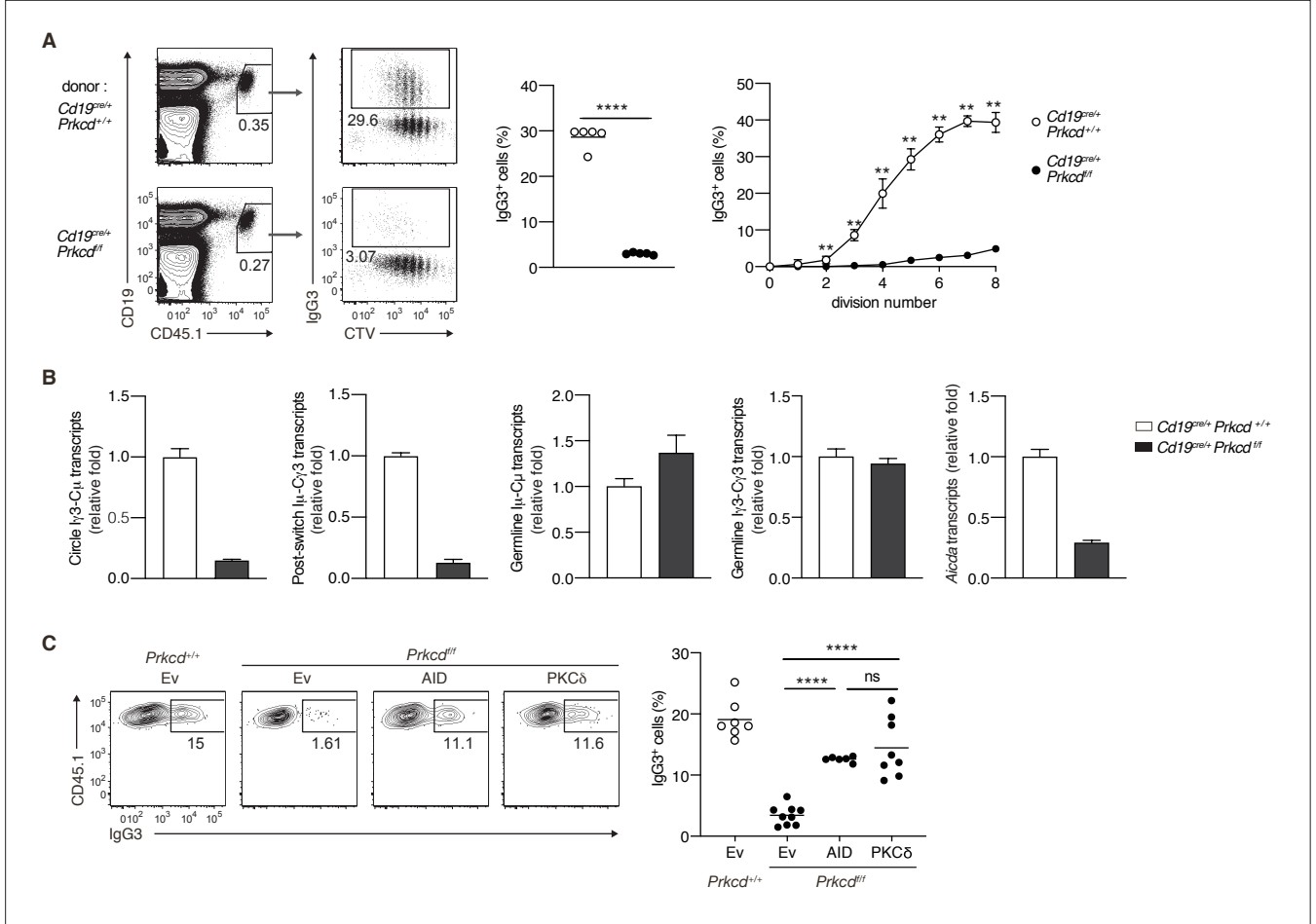

**Figure 4.** Protein kinase C (PKC) δ signaling promotes IgG class switching by inducing activation-induced cytidine deaminase (AID). (**A and B**) B cells purified from *Prkcd⁺/⁺ Cd19^(cre/+) B1-8^(hi)* or *Prkcd^(f/f) Cd19^(cre/+) B1-8^(hi)* mice were labeled with CellTrace Violet (CTV) and transferred into B6 mice, which were immunized with NP-Ficoll on the next day. Spleen cells were analyzed 3 days later. (**A**) Representative flow cytometric plots of the spleen cells with the numbers indicating percentages of the cells within the neighboring gates (left). The frequency of IgG3⁺ cells among whole donor B cells (CD45.1⁺ CD19⁺ CD138⁻) (middle) and at each cell division number (right) are shown. (**B**) qRT-PCR analysis of the indicated transcripts in the donor B cells collected as in *Figure 4—figure supplement 1A*. (**C**) *Prkcd⁺/⁺ Cd19^(cre/+) B1-8^(hi)* or *Prkcd^(f/f) Cd19^(cre/+) B1-8^(hi)* B cells transduced with an empty vector (Ev) or vectors expressing AID or PKC δ were transferred into B6 mice that had been immunized with NP-Ficoll on the previous day as in *Figure 4—figure supplement 1B*. Representative flow cytometry plots of the donor cells transduced with the vectors (CD45.1⁺GFP⁺, gated as in *Figure 4—figure supplement 1C*) (left) and the frequency of IgG3⁺ cells among the CD45.1⁺GFP⁺ cells (right) on day 3 after transfer. Small horizontal bars are the means of five (**A**) or six to nine (**C**) biological replicates. Each symbol represents an individual mouse (A, middle; **C**). The symbols are the means of five biological replicates (A, right). Data are means ± standard deviations (SDs) of three technical replicates pooled from five mice (**B**). The data are representative of three (**A**) or two (**B**) independent experiments or is pooled from two independent experiments (**C**). ns, not significant (p > 0.05); **p < 0.01; ****p < 0.0001; p values were calculated by unpaired multiple *t*-test (**A**) or one-way analysis of variance (ANOVA) with Tukey's test (**C**).

The online version of this article includes the following figure supplement(s) for figure 4:

**Source data 1.** Source data for *Figure 4A–C*.

**Figure supplement 1.** Gating strategies.

It was reported that BATF binds to a regulatory region of the *Aicda* gene to directly promote its expression and that IgG3 production against TNP-Ficoll was impaired in *Batf⁻/⁻* mice (*Ise et al., 2011*). Therefore, we next asked whether the defect of BATF expression is responsible for the suppression of AID expression and IgG3⁺ cell generation in PKCδ-deficient mice. First, we knocked down BATF in *B1-8^(hi)* B cells and transferred them into mice immunized with NP-Ficoll. Both the expression of *Aicda* and the frequency of IgG3⁺ cells were significantly decreased in BATF knockdown cells compared with the mock-transduced control cells (*Figure 5D and E*). Conversely, forced expression of BATF in *B1-8^(hi) Cd19^(cre/+) Prkcd^(f/f)* B cells partially but significantly restored *Aicda* expression and the generation of

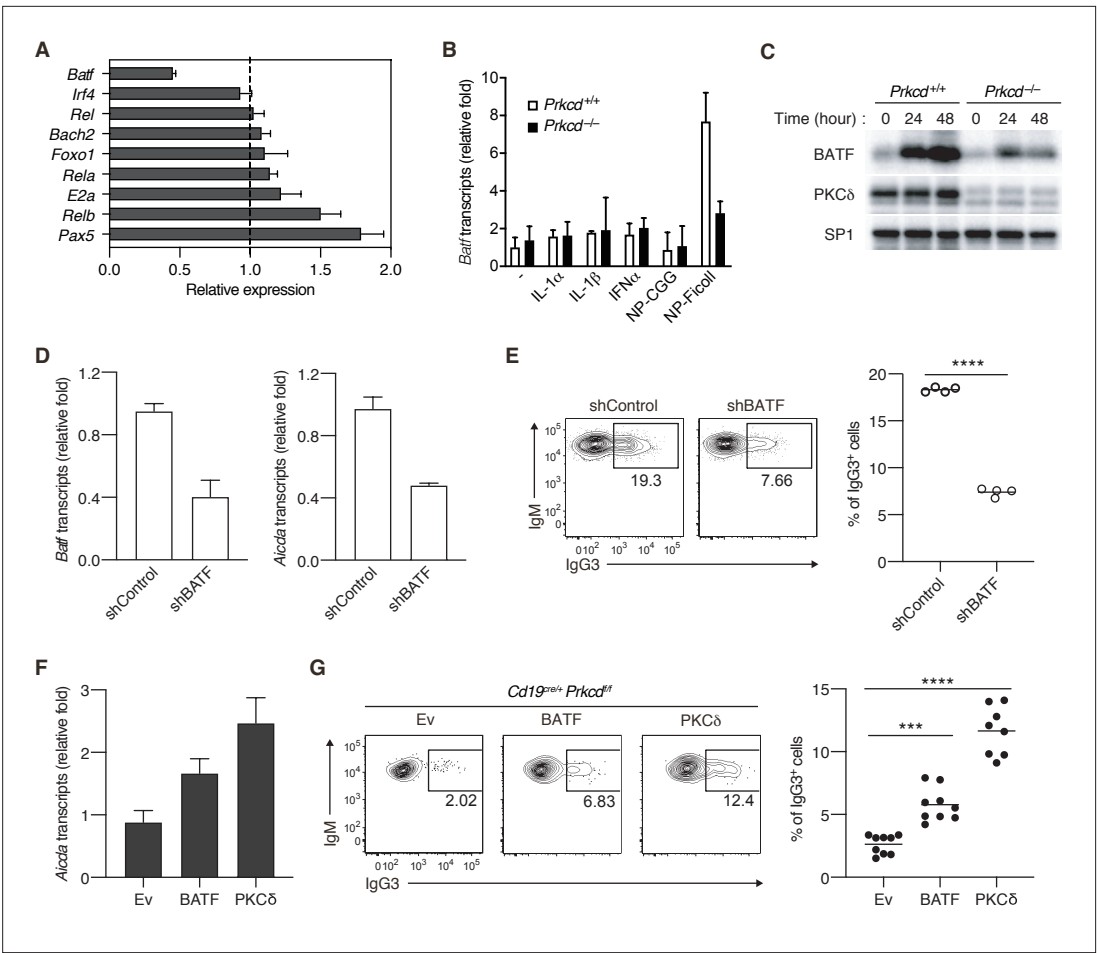

**Figure 5.** Protein kinase C (PKC) δ regulates IgG class switching by upregulating BATF expression. (**A**) qRT-PCR analysis of the transcripts of the indicated genes in *Prkcd⁺/⁺ Cd19ᶜʳᵉ/⁺ B1-8 ʰⁱ* or *Prkcdᶠ/ᶠ Cd19ᶜʳᵉ/⁺ B1-8 ʰⁱ* donor B cells (CD45.1⁺CD19⁺CD138⁻) purified as in *Figure 4—figure supplement 1A* from the recipient mice immunized with NP-Ficoll 3 days previously. Shown is the relative expression of each gene in *Prkcdᶠ/ᶠ Cd19ᶜʳᵉ/⁺* cells to that in *Prkcd⁺/⁺ Cd19ᶜʳᵉ/⁺* cells. (**B**) qRT-PCR analysis of *Batf* transcripts in *Prkcd⁺/⁺ Igk⁻/⁻ B1-8ᶠˡᵒˣ/⁺* or *Prkcd⁻/⁻ Igk⁻/⁻ B1-8ᶠˡᵒˣ/⁺* B cells cultured with medium alone (–) or with the indicated stimuli for 2 days. (**C**) Immunoblot analysis in *Prkcd⁺/⁺ Igk⁻/⁻ B1-8ᶠˡᵒˣ/⁺* or *Prkcd⁻/⁻ Igk⁻/⁻ B1-8ᶠˡᵒˣ/⁺* B cells stimulated with NP-Ficoll for the indicated times. (**D and E**) *B1-8ʰⁱ* B cells transduced with knockdown vectors for luciferase (shControl) or BATF (shBATF) were transferred into B6 mice that had been immunized with NP-Ficoll on the previous day as in *Figure 4—figure supplement 1B*, and their spleen cells were analyzed on day 4 after transfer. (**D**) qRT-PCR analysis of *Aicda* and *Batf* transcripts in the vector-transduced donor B cells (CD45.1⁺ GFP⁺ CD19⁺ CD138⁻) collected as in *Figure 4—figure supplement 1C*. (**E**) Representative flow cytometric plots of the transduced donor cells with the numbers indicating the percentage of IgG3⁺ cells (left) and the frequency of the IgG3⁺ cells among such cells (right) gated as in *Figure 4—figure supplement 1C*. (**F and G**) *Prkcdᶠ/ᶠ Cd19ᶜʳᵉ/⁺ B1-8 ʰⁱ* B cells transduced with Ev or vectors expressing BATF or PKC δ were transferred into B6 mice that had been immunized with NP-Ficoll on the previous day, and their spleen cells were analyzed 3 days after transfer. (**F**) qRT-PCR analysis of *Aicda* transcripts in the vector-transduced donor B cells. (**G**) Representative flow cytometric plots of the transduced donor cells with the numbers indicating the percentage of IgG3⁺ cells (left) and the frequency of the IgG3⁺ cells among such cells (right). Data are means ± standard deviations (SDs) of three technical replicates (**A, B, D, and F**). Samples were pooled from five to eight mice (**D and F**). Small horizontal bars are the means of four (**E**) or eight to nine (**G**) biological replicates. Each symbol represents an individual mouse (**E and G**). The data are representative of two independent experiments (**A–F**) or are pooled from two independent experiments (**G**). ***p < 0.001; ****p < 0.0001; p values were calculated by two-tailed unpaired Student's *t*-test (**E**) or one-way analysis of variance (ANOVA) with Tukey's test (**G**).

The online version of this article includes the following figure supplement(s) for figure 5:

**Source data 1.** Source data for *Figure 5A, B and D–G*.

**Source data 2.** Source data for *Figure 5C*.

**Source data 3.** Source data for *Figure 5C*.

**Source data 4.** Source data for *Figure 5C*.

**Source data 5.** Source data for *Figure 5C*.

*Figure 5 continued on next page*

*Figure 5 continued*

**Figure supplement 1.** The expression of BATF and the phosphorylation of protein kinase C (PKC) δ .

**Figure supplement 1—source data 1.** Source data for *Figure 5—figure supplement 1A*.

**Figure supplement 1—source data 2.** Source data for *Figure 5—figure supplement 1A*.

**Figure supplement 1—source data 3.** Source data for *Figure 5—figure supplement 1A*.

**Figure supplement 1—source data 4.** Source data for *Figure 5—figure supplement 1B*.

**Figure supplement 1—source data 5.** Source data for *Figure 5—figure supplement 1B*.

**Figure supplement 1—source data 6.** Source data for *Figure 5—figure supplement 1B*.

**Figure supplement 1—source data 7.** Source data for *Figure 5—figure supplement 1B*.

**Figure supplement 1—source data 8.** Source data for *Figure 5—figure supplement 1C*.

**Figure supplement 1—source data 9.** Source data for *Figure 5—figure supplement 1C*.

**Figure supplement 1—source data 10.** Source data for *Figure 5—figure supplement 1C*.

**Figure supplement 1—source data 11.** Source data for *Figure 5—figure supplement 1C*.

**Figure supplement 1—source data 12.** Source data for *Figure 5—figure supplement 1D*.

IgG3$^+$ cells in the recipient mice immunized with NP-Ficoll (*Figure 5F and G*). Taken together, these data indicate that PKCδ mediates expression of AID and class switching to IgG3 through upregulation of BATF expression in B cells undergoing a TI-2 response.

## PKCδ is required for homeostatic antibacterial IgG3 production and protection against bacteremia

Recent works have revealed that commensal microbes induce an IgG response and confer protection against systemic bacterial infection (*Zeng et al., 2016*). Among antibacterial IgG, IgG3 is most abundant and produced in a TI manner (*Ansaldo et al., 2019*; *Koch et al., 2016*). Therefore, we assessed the contribution of PKCδ in IgG-mediated antibacterial responses. To standardize the microbiota, we cohoused control and *Cd19$^{cre/+}$ Prkcd$^{f/f}$* mice over 4 weeks and serum antibodies against fecal bacteria were titrated. Production of serum antibacterial IgM, IgG1, and IgG2b was not changed significantly, but that of IgG3 was severely impaired in *Cd19$^{cre/+}$ Prkcd$^{f/f}$* mice (*Figure 6A*). Antibacterial IgG2c was undetectable in both mouse groups (data not shown). Given that PKCδ was required for production of all IgG subclasses in the anti-NP TI-2 response (*Figure 3*), the PKCδ-independent production of antibacterial IgG1 and IgG2b may be attributable to TD responses, as reported for IgG1 (*Ansaldo et al., 2019*), while antibacterial IgG3 is mainly produced by the TI-2 response.

We further asked whether regulation of commensal bacteria is defective in the PKCδ-deficient mice. Dextran sodium sulfate (DSS) treatment is known to disrupt the gut epithelium and to allow intestinal bacteria to translocate throughout the body. Subsequently, it leads to fatal bacteremia in the absence of microbiota-specific IgG (*Zeng et al., 2016*). After the treatment with DSS, *Cd19$^{cre/+}$ Prkcd$^{f/f}$* mice exhibited increased numbers of aerobic and anaerobic bacteria in the blood compared to control mice (*Figure 6B*). Accordingly, the mortality of *Cd19$^{cre/+}$ Prkcd$^{f/f}$* mice was significantly higher than that of control mice (*Figure 6C*). Collectively, these results suggest that IgG3 production by a TI-2 response via PKCδ prevents lethal bacteremia.

## Discussion

Although previous reports showed that proximal BCR signal molecules such as BLNK or Btk are necessary for B-cell activation and subsequent IgM and IgG production in the TI-2 response in mice (*Xu et al., 2000*; *Ellmeier et al., 2000*), it was not clear whether there are any specific BCR signaling pathways inducing CSR. Here, we demonstrated that BCR stimulation with a TI-2 Ag (NP-Ficoll), but not a TD Ag (NP-CGG), both sharing the same BCR epitope, promoted transcription of *Aicda* and induced IgG3 CSR in the presence of a secondary stimulation. BCR engagement with a TI-2 Ag induced the phosphorylation of PKCδ and PKCδ was required for upregulation of BATF expression, thereby mediated the induction of *Aicda* transcription and subsequent CSR to the IgG subclasses (*Figure 7*). By contrast, PKCδ was dispensable for TI-2 Ag-mediated B-cell proliferation, IgM production, as well as

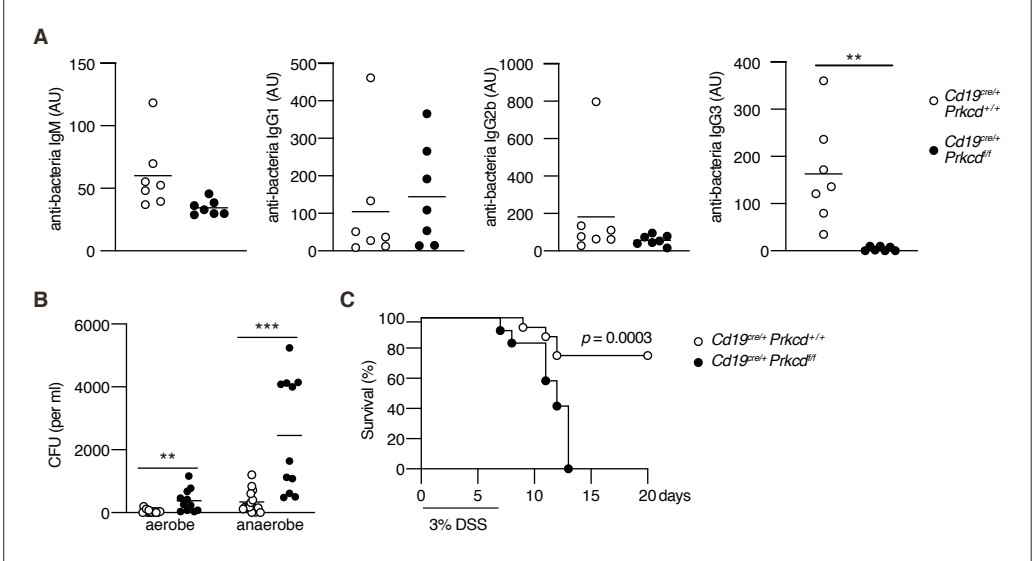

**Figure 6.** Protein kinase C (PKC) δ is required for protective antibacterial IgG3 production. *Cd19*[cre/+] *Prkcd*[+/+] and *Cd19*[cre/+] *Prkcd*[f/f] mice were cohoused for at least 4 weeks and treated with 3 % dextran sodium sulfate (DSS) for 7 days. (**A**) Serum IgM, IgG1, IgG2b, and IgG3 against fecal bacteria after cohousing were quantified by enzyme-linked immunosorbent assay (ELISA). AU, arbitrary units. (**B**) Colony-forming unit (CFU) of aerobes and anaerobes in the blood 7 days after DSS treatment. (**C**) Kaplan–Meier survival plot of 16 (*Cd19*[cre/+] *Prkcd*[+/+]) and 12 (*Cd19*[cre/+] *Prkcd*[f/f]) mice at indicated days. Small horizontal bars are the means of 7 (**A**), or 15 (*Cd19*[cre/+] *Prkcd*[+/+]) or 11 (*Cd19*[cre/+] *Prkcd*[f/f]) in (**B**), biological replicates. Data were obtained from 16 (*Cd19*[cre/+] *Prkcd*[+/+]) or 12 (*Cd19*[cre/+] *Prkcd*[f/f]) mice in (**C**). Each symbol represents an individual mouse (**A and B**). The data are representative of two independent experiments (**A**) or pooled from two independent experiments (**B and C**). **p < 0.01; ***p < 0.001; p values were calculated by two-tailed unpaired Welch's *t*-test (**A**), unpaired multiple *t*-test (**B**), or log-rank test (**C**).

The online version of this article includes the following figure supplement(s) for figure 6:

**Source data 1.** Source data for *Figure 6A and B*.

for B-cell development and the TD response. Thus, we have found for the first time, as far as we know, a BCR signaling molecule that is selectively required for CSR in the TI-2 response.

TI-2 Ags such as NP-Ficoll can induce antibody response in the absence of major histocompatibility complex class II-restricted T cell help or MyD88-mediated TLR receptor signaling (*Gavin et al., 2006*; *Hou et al., 2011*). Thus, it has been unclear whether any costimulation is necessary for B-cell activation in a TI-2 response. We demonstrated that NP-Ficoll, but not NP-CGG, was able to activate NP-specific naive B cells to induce proliferation, antibody production, and potentiation for CSR in vitro. Thus, despite sharing the same antigenic epitope (NP) at a similar average number per molecule, these Ags appear to elicit disparate BCR signal transduction. After Ag binding, the BCR is clustered and recruited to lipid microdomains, in which signaling proteins are localized that trigger B-cell activation signaling (*Niiro and Clark, 2002*). Ag structure probably affects such BCR dynamics or membrane organization of signal components. Consistently, it was previously reported that a high-valency TI-2 Ag induces the formation of large BCR clusters in the lipid microdomains (*Puffer et al., 2007*).

We have revealed that, in the TI-2 response, BCR-downstream signaling through PKCδ is critical for CSR and generation of IgG. PKCδ is known to control Ag-induced tolerance in immature B cells (*Mecklenbräuker et al., 2002*; *Limnander et al., 2011*). However, a possible alteration of the mature B-cell repertoire in PKCδ-deficient mice is not attributable to the defective class switching to IgG3, since the defect was obvious in the NP-Ficoll response by PKCδ-deficient, B1-8V$_H$-knock-in B cells with a monoclonal anti-NP repertoire. Here, we demonstrated that PKCδ mediated the expression of AID to induce CSR. BCR engagement with a TI-2 Ag induced the expression of BATF via PKCδ and BATF was required for the transcription of the *Aicda* gene in a TI-2 response, as reported previously (*Ise et al., 2011*). Forced expression of BATF in PKCδ-deficient B cells restored the expression of AID and IgG3 class switching significantly, but less effectively, than that of PKCδ itself. In this regard, various transcription factors have been reported to cooperatively induce the transcription

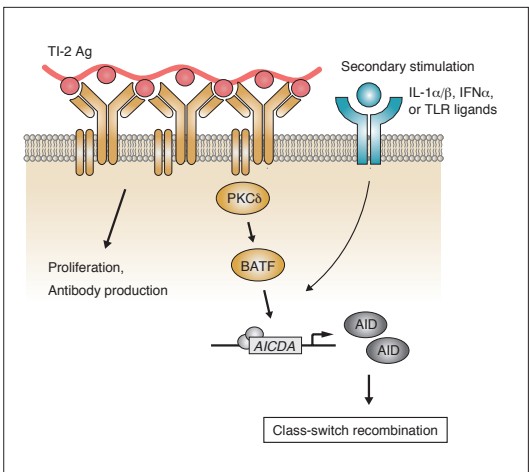

**Figure 7.** Working model of the protein kinase C (PKC) δ -mediated induction of class-switch recombination (CSR) in the T-cell-independent type-2 (TI-2) response. B-cell receptor (BCR) stimulation with a TI-2 antigen (Ag) activates PKC δ signaling and induces the expression of BATF, which works cooperatively with secondary signals induced by IL-1α/β, IFNα, or TLR ligands and drives the transcription of activation-induced cytidine deaminase (AID) to induce IgG CSR.

of AID in the TD response (**Vaidyanathan et al., 2014**; **Crouch et al., 2007**; **Tran et al., 2010**). Although we have not assessed the contribution of other transcription factors in the TI-2 response, since the expression of other candidate genes was intact in PKCδ-deficient B cells (**Figure 5A**), PKCδ signaling might regulate other transcription factors in addition to BATF to fully upregulate the expression of AID to induce CSR in the TI-2 response.

PKCδ contains several phosphorylation sites, whose phosphorylation pattern affects its enzymatic activity and possibly the selection of downstream targets (**Salzer et al., 2016**; **Steinberg, 2004**). Although the phosphorylation of Thr 505 and Ser 647 of PKCδ is known to be important for its enzymatic activity, phosphorylation of these sites was constitutive in unstimulated naive B cells and did not increase after NP-Ficoll stimulation (data not shown). This observation and a previous report that PKCδ exerts proapoptotic activity in peripheral B cells through nuclear translocation, unless BAFF signaling reverses it (**Mecklenbräuker et al., 2004**), indicate a constitutive PKCδ activity in B cells irrespective of Ag stimulation. Augmented IgM production after immunization with TI-2 as well as TD Ags in PKCδ-deficient mice (**Figure 3A and D** and **Figure 3—figure supplement 1B**) may reflect the lack of the general proapoptotic function of PKCδ. On the other hand, phosphorylation of PKCδ at Tyr 311, shown here to be selectively augmented by TI-2 Ag stimulation, is known to alter its intracellular localization and the substrate specificity (**Rybin et al., 2004**; **Steinberg, 2004**) and to be involved in various cellular responses (**Balasubramanian et al., 2006**; **Nakashima et al., 2008**). Thus, to understand the mechanism of TI-2 Ag-specific BCR signaling that induces the expression of BATF and AID, it would be necessary to identify a specific substrate of the phosphorylated PKCδ at Tyr 311.

In contrast to TI-2 response, PKCδ was not required for IgG production in the TD response, like other proximal BCR molecules (**Khan et al., 1995**; **Xu et al., 2000**). In vitro stimulation with a TD Ag alone induced only a marginal level of phosphorylation of PKCδ and other signaling molecules (**Figure 2A** and data not shown) and did not induce proliferation, IgM production, and the expression of BATF. Accordingly, we assume that the TD Ag-mediated BCR signaling cannot exceed an activation threshold to induce gene expression leading to functional B-cell response including class switching. Instead, other stimuli such as CD40L would meet the demand for B-cell activation in TD response. Supporting this view, it was reported that BCR signaling does not enhance the class switching induced by CD40 signaling (**Pone et al., 2012**).

The synergistic effect of TLR and BCR on the induction of AID has been shown previously (**Pone et al., 2012**). Here, we found that IL-1α/β, IFNα, and TLR ligands costimulate B cells with TI-2 Ag to induce *Aicda* expression and class switching to IgG3 and perhaps other IgG subclasses, while other cytokines tested barely induced IgG3 production in vitro. In the context of TD response, the recruitment of NF-$\kappa$B and STAT6 to the 5′ upstream enhancer region of *Aicda* locus, in addition to BATF recruitment to the 3′ downstream region, is required for the transcription of *Aicda* (**Tran et al., 2010**; **Vaidyanathan et al., 2014**). Since the in vitro stimulation by TI-2 Ag induced the expression of BATF but not of *Aicda* without the additional stimulation, such costimulation may signal the recruitment of transcription factors to the 5′ upstream enhancer region. In this regard, it was reported that IL-1 and IFNα induce the transcription of *Aicda* through activation of NF-$\kappa$B in hepatocytes during hepatitis B virus infection (**Watashi et al., 2013**). IL-1 and IFNα have also been reported to activate STAT proteins,

if not STAT6 (*Biffi et al., 2019*; *Du et al., 2007*). However, further study would be necessary to understand the overall mechanism for the induction of *Aicda* expression in B cells in the TI-2 response.

Consistent with our in vitro data, previous reports showed the contribution of IL-1α/β, IFNα, and TLR ligands in several types of TI-2 responses: IgG production upon immunization with NP-Ficoll was dampened in IL-1α/β double-deficient mice (*Nakae et al., 2001*), and IgG2c production with NP-Ficoll plus poly(I:C) was dependent on the IFNα receptor on B cells (*Swanson et al., 2010*). While IgG production against NP-Ficoll did not require MyD88-mediated TLR signaling (*Gavin et al., 2006*; *Hou et al., 2011*), IgG production against pneumococcal polysaccharides requires TLR signaling (*Sen et al., 2005*). Compared to NP-Ficoll, physiological TI-2 Ags are more complex (*Snapper, 2006*) and may contain pathogen-associated molecular patterns (PAMPs) that mediate TLR signaling. Therefore, any of IL-1α/β, IFNα, and TLR ligands could function as costimulation for an in vivo TI-2 response, depending on the type of Ag.

Our data revealed that PKCδ is generally required for IgG production in response to TI-2 Ags such as NP-Ficoll and PPS3. Given the predominant role of IgG3 in antipneumococcal responses (*McLay et al., 2002*), PKCδ seems to be needed for protection from pneumococcal infection. Furthermore, we found that PKCδ is required to produce IgG3 against commensal bacteria in the steady state and the prevention of bacteremia after epithelial barrier disruption. As commensal bacteria have been shown to contain various Ags and stimulate multiple immune pathways (*Belkaid and Harrison, 2017*), our result suggests that some commensal bacteria express TI-2 Ag-like repetitive structures that elicit IgG3 production. It has been shown that such IgG against symbiotic bacteria also plays a protective role in systemic infection by pathogens (*Zeng et al., 2016*). Taken together, IgG production in the TI-2 response seems to be critical for regulation of various types of bacteria.

Several reports uncovered the potential contribution of the dysregulated microbiome in SLE pathogenesis together with genetic risk factors (*Silverman, 2019*). Translocation of a gut pathobiont to the liver and other systemic tissues causes a lupus-like disease in genetically autoimmune-prone mice owing to induction of a systemic type I interferon response and autoantibody production (*Manfredo Vieira et al., 2018*). Prevention of this bacterial translocation and autoimmune response by vaccination against the pathobiont indicated a role of antibodies for the prevention. Thus, besides the defect of B-cell tolerance (*Miyamoto et al., 2002*; *Mecklenbräuker et al., 2002*), the low serum antibacterial IgG3 in PKCδ-deficient mice may lead to the pathobiont invasion and autoantibody production through intact TD response. Loss-of-function PKCδ mutations in humans also cause SLE-like autoimmunity (*Salzer et al., 2016*). Some of these patients show reduced IgG-positive B cells in the peripheral blood and have recurrent infections for unknown reasons (*Kiykim et al., 2015*; *Salzer et al., 2013*; *Kuehn et al., 2013*). Taken together, PKCδ-mediated IgG production by the TI-2 response appears to be critical also in humans for host defense against certain bacteria and the regulation of autoimmunity.

# Materials and methods

**Key resources table**

| Reagent type (species) or resource | Designation | Source or reference | Identifiers | Additional information |
|---|---|---|---|---|
| Strain, strain background (*M. musculus*) | C57BL/6NCrSlc (B6) | Japan SLC Inc | | |
| Genetic reagent (*M. musculus*) | C57BL/6 CD45.1 | RIKEN BRC | RBRC00144 | |
| Genetic reagent (*M. musculus*) | B1-8^flox/+ | doi: 10.1016/S0092-8674(00)80373–6. | | |
| Genetic reagent (*M. musculus*) | B1-8^hi | doi:10.1038/ni776. | JAX stock #007775 | |
| Genetic reagent (*M. musculus*) | Cd19^cre/+ | doi:10.1038/376352a0 . | JAX stock #006785 | |
| Genetic reagent (*M. musculus*) | Prkcd^−/− | RIKEN BRC | RBRC00457 | |

*Continued on next page*

*Continued*

| Reagent type (species) or resource | Designation | Source or reference | Identifiers | Additional information |
|---|---|---|---|---|
| Genetic reagent (*M. musculus*) | *Prkcd*[f/f] | RIKEN BRC | RBRC06462 | |
| Genetic reagent (*M. musculus*) | *Igk*[−/−] | doi: 10.1002/j.1460–2075.1993.tb05722.x. | | |
| Antibody | APC-Cy7 anti-mouse CD19 (Rat monoclonal) | BioLegend | Cat# 115,530 | Flow cytometry (1/200) |
| Antibody | Biotin Rat Anti-Mouse CD43 (Rat monoclonal) | BD Biosciences | Cat# 20,737 | Flow cytometry (1/300) |
| Antibody | BV421 anti-mouse CD138 (Rat monoclonal) | BioLegend | Cat# 142,508 | Flow cytometry (1/500) |
| Antibody | BV421 Rat Anti-Mouse IgG3 (Rat monoclonal) | BD Biosciences | Cat# 565,808 | Flow cytometry (1/300) |
| Antibody | CD45.1 Monoclonal Antibody (A20), APC (Mouse monoclonal) | Invitrogen | Cat# 17-0453-82 | Flow cytometry (1/200) |
| Antibody | FITC anti-mouse CD21/CD35 (CR2/CR1) Antibody (Rat monoclonal) | BioLegend | Cat# 123,407 | Flow cytometry (1/300) |
| Antibody | FITC anti-mouse CD5 Antibody (Rat monoclonal) | BioLegend | Cat# 100,605 | Flow cytometry (1/300) |
| Antibody | Goat Anti-Mouse IgG2c, Human ads-FITC (Goat Polyclonal) | Southern Biotech | Cat# 1079-02 | Flow cytometry (1/300) |
| Antibody | Goat F(ab')2 Anti-Mouse IgG2b, Human ads-FITC (Goat Polyclonal) | Southern Biotech | Cat# 1092-02 | Flow cytometry (1/300) |
| Antibody | Goat F(ab')2 Anti-Mouse IgG3, Human ads-FITC (Goat Polyclonal) | Southern Biotech | Cat# 1102-02 | Flow cytometry (1/300) |
| Antibody | PE anti-mouse CD23 Antibody (Rat monoclonal) | BioLegend | Cat# 101,607 | Flow cytometry (1/300) |
| Antibody | PE anti-mouse IgG2b (Rat monoclonal) | BioLegend | Cat# 406,707 | Flow cytometry (1/500) |
| Antibody | PE-Cy7 anti-mouse CD19 (Rat monoclonal) | BioLegend | Cat# 115,520 | Flow cytometry (1/200) |
| Antibody | Goat Anti-Mouse IgG Fc-HRP (Goat Polyclonal) | SouthernBiotech | Cat# 1033-05 | ELISA (1/2000) |
| Antibody | Goat Anti-Mouse IgG, Human ads-UNLB (Goat Polyclonal) | SouthernBiotech | Cat# 1030-01 | ELISA (1/1000) |
| Antibody | Goat Anti-Mouse IgG1, Human ads-HRP (Goat Polyclonal) | SouthernBiotech | Cat# 1070-05 | ELISA (1/2000) |
| Antibody | Goat Anti-Mouse IgG1, Human ads-UNLB (Goat Polyclonal) | SouthernBiotech | Cat# 1070-01 | ELISA (1/1000) |
| Antibody | Goat Anti-Mouse IgG2b, Human ads-HRP (Goat Polyclonal) | SouthernBiotech | Cat# 1090-05 | ELISA (1/2000) |

*Continued on next page*

*Continued*

| Reagent type (species) or resource | Designation | Source or reference | Identifiers | Additional information |
|---|---|---|---|---|
| Antibody | Goat Anti-Mouse IgG2b, Human ads-UNLB (Goat Polyclonal) | SouthernBiotech | Cat# 1090-01 | ELISA (1/1000) |
| Antibody | Goat Anti-Mouse IgG2c, Human ads-HRP (Goat Polyclonal) | SouthernBiotech | Cat# 1079-05 | ELISA (1/2000) |
| Antibody | Goat Anti-Mouse IgG3, Human ads-UNLB (Goat Polyclonal) | SouthernBiotech | Cat# 1100-01 | ELISA (1/1000) |
| Antibody | Goat Anti-Mouse IgG3, Human/Bovine/Horse SP ads-HRP (Goat Polyclonal) | SouthernBiotech | Cat# 1103-05 | ELISA (1/2000) |
| Antibody | Goat Anti-Mouse IgM, Human ads-HRP (Goat Polyclonal) | SouthernBiotech | Cat# 1020-05 | ELISA (1/2000) |
| Antibody | Goat Anti-Mouse IgM, Human ads-UNLB (Goat Polyclonal) | SouthernBiotech | Cat# 1020-01 | ELISA (1/1000) |
| Antibody | Anti-GAPDH mAb (Mouse monoclonal) | MBL | Cat# M171-3 | WB (1/3000) |
| Antibody | Anti-SP1 antibody (Rabbit polyclonal) | Abcam | Cat# ab227383 | WB (1/5000) |
| Antibody | BATF (D7C5) Rabbit mAb (Rabbit monoclonal) | Cell Signaling Technology | Cat# 8638 | WB (1/2000) |
| Antibody | Phospho-PKCdelta (Tyr311) Antibody (Rabbit polyclonal) | Cell Signaling Technology | Cat# 2055 | WB (1/2000) |
| Antibody | PKC δ Antibody (C-17) (Rabbit polyclonal) | Santa Cruz Biotechnology | Cat# sc-213 | WB (1/2000) |
| Recombinant DNA reagent | pMXs-IRES-GFP | other | | Dr. Kitamura (University of Tokyo) |
| Recombinant DNA reagent | pMXs-BATF-IRES-GFP | This paper | | BATF (*M. musculus*) |
| Recombinant DNA reagent | pMXs-AID-IRES-GFP | This paper | | AID (*M. musculus*) |
| Recombinant DNA reagent | pMXs-PKCδ-IRES-GFP | This paper | | PKCδ (*M. musculus*) |
| Recombinant DNA reagent | pSIREN-GFP shLuciferase (shControl) | doi: 10.1016/j.celrep.2020.108333. | | |
| Recombinant DNA reagent | pSIREN-GFP shBATF | This paper | | Taget sequence (5′–3′): ACCCTGGACTGTCATGAATGA |
| Recombinant DNA reagent | pVSV-G | Clontech | Cat# 631,530 | |
| Sequence-based reagent | Primers | This paper | | See *Supplementary file 1* |
| Sequence-based reagent | Oligo(dT) 20 Primer | Thermo Fisher Scientific | Cat# 18418020 | |
| Peptide, recombinant protein | NP46-Ficoll | Biosearch Technologies | Cat# F-1420 | |
| Peptide, recombinant protein | NP40-CGG | doi:10.1038/ni.3508. | | |
| Peptide, recombinant protein | NP14-BSA-Alexa Fluor 647 | doi:10.1038/ni.3508. | | |
| Peptide, recombinant protein | Cytokines | See *Supplementary file 1* | | |

*Continued on next page*

*Continued*

| Reagent type (species) or resource | Designation | Source or reference | Identifiers | Additional information |
|---|---|---|---|---|
| Commercial assay or kit | Fixation/ Permeabilization Solution Kit | BD Biosciences | Cat# 554,714 | |
| Commercial assay or kit | QIAquick PCR Purification Kit | QIAGEN | Cat# 28,106 | |
| Commercial assay or kit | RNeasy Micro | QIAGEN | Cat# 74,004 | |
| Commercial assay or kit | Thunderbird SYBR qPCR Mix | TOYOBO | Cat# QPS-201 | |
| Commercial assay or kit | SuperPrep II Cell lysis & RT Kit for qPCR | TOYOBO | Cat# SCQ-401 | |
| Commercial assay or kit | CellTraceTM Violet (CTV) Cell proliferation Kit | Thermo Fisher Scientific | Cat# C34557 | |
| Chemical compound, drug | PPS3 | ATCC | Cat# 169 X | |
| Chemical compound, drug | LPS | Sigma | Cat# L2880 | |
| Chemical compound, drug | R-848 | InvivoGen | Cat# tlrl-r848 | |
| Chemical compound, drug | CpG ODN 1826 | InvivoGen | Cat# tlrl-1826 | |
| Chemical compound, drug | Fixable Viability Dye eFluor 506 | Thermo Fisher Scientific | Cat# 65-0866-18 | |
| Chemical compound, drug | DOTAP Liposomal Transfection Reagent | Sigma | Cat# 11202375001 | |
| Chemical compound, drug | TRI Reagent | Sigma | Cat# T9424 | |
| Chemical compound, drug | PEI Max (Mw 40,000) | Polysciences | Cat# 24765-1 | |
| Chemical compound, drug | ReverTra Ace | TOYOBO | Cat# TRT-101 | |
| Chemical compound, drug | KOD Fx Neo DNA polymerase | TOYOBO | Cat# KFX-201 | |
| Chemical compound, drug | [$^3$ H] thymidine | PerkinElmer | Cat# NET027001MC | |
| Chemical compound, drug | anti-APC MicroBeads | MicroBeads | Cat# 130-090-855 | |
| Chemical compound, drug | GoTaq Green Master Mix | Promega | Cat# M712B | |
| Chemical compound, drug | DSS | MP Biomedicals | Cat# 9011-18-1 | |
| Chemical compound, drug | Brain Heart Infusion Agar | BD Bioscience | Cat# 221,570 | |
| Software, algorithm | FlowJo | https://www.flowjo.com/solutions/flowjo | RRID:SCR_008520 | |
| Software, algorithm | GraphPad Prism | https://graphpad.com | RRID:SCR_002798 | |

## Mice and immunizations

C57BL/6NCrSlc (B6) mice were purchased from Japan SLC. All the following mice were backcrossed to the B6 or B6-CD45.1 strain: $B1-8^{flox/+}$ mice (**Lam et al., 1997**), $B1-8^{hi}$ mice (**Shih et al., 2002**), $Igk^{-/-}$ mice (**Chen et al., 1993**), $Cd19^{cre/+}$ mice (**Rickert et al., 1995**), and $Prkcd^{-/-}$ mice (**Miyamoto et al., 2002**). $Prkcd^{fl/fl}$ mice on the B6 background were developed by Drs. Niino, Shioda, and Sakimura (**Niino et al., 2021**) and purchased from the RIKEN BioResource Center (RBRC06462). Mice were immunized i.p. with 100 µg of $NP_{46}$-Ficoll (F-1420; Biosearch Technologies), 1 µg of PPS3 (169 X; American Type Culture Collection) or 100 µg of $NP_{40}$-CGG in alum (**Haniuda et al., 2016**) unless otherwise noted. For flow cytometry of spleen cells, mice were immunized i.v. with 100 µg of $NP_{46}$-Ficoll. Sex-matched 7–14- week-old mice were used for all experiments. All mice were maintained in the Tokyo University of Science (TUS) mouse facility under specific pathogen-free conditions. Mouse procedures were performed under protocols approved by the TUS Animal Care and Use Committee (Approval No. S19017 and S20011).

## B-cell culture

Spleen cells were stained with a cocktail of biotinylated Abs for CD4, CD8, CD43, CD49b, Ter119, and Streptavidin Particles Plus DM, from which naive B cells were purified by magnetic negative sorting using the IMag system (BD Biosciences) and MACS system (Miltenyi Biotec), as described previously (**Nojima et al., 2011**). B cells were cultured in RPMI-1640 medium (Wako) supplemented with 10 % heat-inactivated fetal bovine serum, 10 mM HEPES pH 7.5, 1 mM sodium pyruvate, 50 mM 2-mercaptoethanol, 100 U/ml penicillin, and 100 mg/ml streptomycin (GIBCO). Typically, B cells were cultured at $2 \times 10^5$ /ml in the presence of the following stimuli at the indicated doses, unless otherwise noted: $NP_{46}$-Ficoll (10 ng/ml), $NP_{40}$-CGG (10 ng/ml), LPS (1 µg/ml, L2880; Sigma), R-848 (1 µg/ml, tlrl-r848; InvivoGen), CpG ODN 1826 (1 µg/ml, tlrl-1826; InvivoGen), IL-1α (1 ng/ml, 211–11 A; Pepro Tech), IL-1β (1 ng/ml, 211-11B; Pepro Tech), or IFNα (100 ng/ml, 752802; Biolegend). Concentrations of cytokines used in **Figure 1—figure supplement 1D** are shown in **Supplementary file 1**.

## Retroviral transduction

To produce retrovirus, pSIREN- or pMXs-based plasmids were cotransfected together with pVSVG into Plat-E cells (kindly provided by T. Kitamura, University of Tokyo) using PEI Max (Mw 40,000, 24765-1; Polysciences). The virus-containing supernatant was harvested 2 days after transfection. For retroviral transduction, B cells were preactivated in vivo: $B1-8^{hi}$ mice were injected i.p. with 50 µg of NP-Ficoll, and then B cells were purified from the spleens of these mice on the next day. These B cells were mixed with the virus-containing supernatant and spin infected at 2000 rpm, 37 °C for 90 min with 10 mg/ml DOTAP Liposomal Transfection Reagent (11202375001; Sigma). One day later, the cells were harvested and $5 \times 10^5$ cells were transferred into B6 mice that had been immunized i.v. with NP-Ficoll on the previous day. This strategy is summarized in **Figure 4—figure supplement 1B**.

## Proliferation assay

The proliferation assay was performed as described previously (**Fukao et al., 2014**). Naive B cells were cultured at $5 \times 10^4$ cells/well in 96-well plates for 72 hr, with the last 8 hr in the presence of [$^3$H] thymidine (1 mCi/well, NET027001MC; PerkinElmer). Incorporated [$^3$H] thymidine was counted by a BetaPlate scintillation counter (Wallac, Gaithersburg, MD).

## Flow cytometry

Single-cell suspensions from spleen or peritoneal cavity were prepared, red blood cells were lysed with ammonium chloride buffer and then cells were incubated with anti-CD16/32 Ab (2.4G2) to block FcγRs. Cultured B cells were collected in MACS buffer (phosphate-buffered saline [PBS] supplemented with 0.5 % BSA, 2 mM ethylenediaminetetraacetic acid [EDTA]) at the indicated days of culture. Cells were stained with Abs and reagents on ice (for splenocytes) or at room temperature (for cultured B cells). For the staining of IgM, IgG, and NP-binding Ig, Fixation/Permeabilization Solution Kit (554714; BD Biosciences) was used according to the manufacturer's protocol to detect total (surface and intracellular) proteins, after surface staining of other molecules. NP-binding Ig was stained with $NP_{14}$-BSA-Alexa Fluor 647 (**Haniuda et al., 2016**). Dead cells were stained with Fixable Viability dye eFluor 506 (65-0866-18; Thermo Fisher Scientific) before cell fixation and excluded from analysis. All samples

were analyzed using FACSCanto II, FACSAria II or III (BD Biosciences) with FlowJo software (Tree Star, Inc).

## Cell division analysis

Naïve B cells were resuspended in PBS at $5 \times 10^6$ cells/ml and labeled with 5 µM of CTV Cell proliferation Kit (C34557; Thermo Fisher Scientific) at 37 °C for 20 min according to the manufacturer's protocol. Collected cells were analyzed by flow cytometry as described above. Cell divisions were determined using the proliferation platform of FlowJo.

## Adaptive transfer and donor B-cell purification

$1 \times 10^6$ CD45.1 *B1-8^{hi}* naive B cells were transferred into B6 mice, which were then immunized i.v. with NP-Ficoll on the next day. Donor B cells were magnetically enriched from pooled spleens of recipient mice using APC-conjugated anti-CD45.1 and anti-APC MicroBeads (130-090-855; Miltenyi Biotec), with the MACS system (Miltenyi Biotec). After enrichment, cells were further stained with respective Abs and sorted using FACSAria II or III (BD Biosciences) as shown in *Figure 4—figure supplement 1A*. Rv-transduced donor B cells were enriched as described above and sorted as shown in *Figure 4— figure supplement 1C*.

## RT-PCR and qPCR

TRI Reagent (T9424; Sigma) or RNeasy Micro (74004; QIAGEN) was used to isolate total RNA from B cells. cDNA was generated from total RNA using ReverTra Ace (TRT-101; TOYOBO) with an oligo(dT)20 primer (18418020; Thermo Fisher Scientific) according to the manufacturer's protocols. For the analysis of a small number of cells, cDNA was generated from cell lysates with SuperPrep II Cell lysis & RT Kit for qPCR (SCQ-401; TOYOBO) according to the manufacturer's protocols. Quantitative real-time PCR (qPCR) was performed using Thunderbird SYBR qPCR Mix (QPS-201; TOYOBO) with the 7500 fast Real-time PCR system or Quant-Studio 3 (Applied Biosystems). For quantification of gene expression levels, each sample was normalized to the expression of a control housekeeping gene, *Gapdh* or *Rps18*. The relative fold change in expression of each gene compared to a control sample, set as 1.0, was calculated with the 2-ddCT method. Primers used in this study are listed in *Supplementary file 1*. The germline and postswitched transcripts were analyzed with the following primer sets: germline Iµ-Cµ transcripts: Iµ Fw1 and Cµ Rv; Iγ3-Cγ3 transcripts: inner Iγ3 Fw and Cγ3 Rv; postswitched Iµ-Cγ3 transcripts: Iµ Fw2 and Cγ3 Rv. For the measurement of the circle Iγ3-Cµ transcript, cDNA was generated with external Cµ Rv primer from total RNA as described above, and preamplified with external Iγ3 Fw primer and Cµ Rv primer using GoTaq Green Master Mix (M712B; Promega). Preapplication products were purified with QIAquick PCR Purification Kit (28104; QIAGEN) and circle Iγ3-Cµ transcript was quantified with inner Iγ3 Fw primer and Cµ Rv primer by qPCR as described above. Specific amplification of the circle Iγ3-Cµ transcript was confirmed by analyzing the sequence of the qPCR products in advance. The expression of the Iγ3-Cµ transcript was normalized to the expression of Rps18 in cDNA generated with oligo(dT)20 primer from the same RNA sample.

## Enzyme-linked immunosorbent assay

Concentrations of total IgM, IgG, or IgG subclasses and of Ag-specific Igs (where indicated) were assessed by titration of culture supernatants or sera by enzyme-linked immunosorbent assay (ELISA). Total and NP-specific antibodies were measured as described previously (*Fukao et al., 2014*; *Nojima et al., 2011*), with NP$_{13.6}$-BSA used for coating plates for the latter. PPS3-specific IgM and IgG3 were detected using 96-well plates coated with PPS3. Bacteria-specific antibody was detected as described previously (*Zeng et al., 2016*). Heat-killed fecal bacteria isolated from *Prkcd^{+/+} Cd19^{cre/+}*mice and *Prkcd^{f/f} Cd19^{cre/+}*mice cohoused at least for 4 weeks were mixed and used for plate coating.

## Immunoblotting

Cells were lysed with 1% NP-40 lysis buffer or RIPA buffer (40 mM Tris–HCl pH 7.5, 150 mM NaCl, 1% NP-40, 1 % sodium deoxycholate, 0.1 % SDS, and 1 mM EDTA) supplemented with protease and phosphatase inhibitors. Lysates were sonicated and mixed with sample buffer and dithiothreitol and boiled. Lysates were resolved on sodium dodecyl sulfate-polyacrylamide gel electrophoresis and

transferred to polyvinylidene fluoride (PVDF) membranes (Millipore), followed by immunoblotting as previously described (*Haniuda et al., 2020*).

## DSS-induced bacteremia

*Prkcd*$^{+/+}$ *Cd19*$^{cre/+}$ miceand *Prkcd*$^{f/f}$ *Cd19*$^{cre/+}$ female mice were cohoused at 1:1 ratio from 4 weeks of age. After at least 4 weeks, cohoused mice were administered 3 % DSS (9011-18-1; MP Biomedicals) in drinking water for 7 days and then switched to regular water. On the day of the last DSS administration, blood samples were collected from mice and cultured on Brain Heart Infusion Agar (221570; BD Bioscience) under aerobic or anaerobic conditions for 24 hr to measure CFU.

## Statistical analysis

Biological replication is derived from multiple biological samples (mouse or cell). Technical replication consisted of multiple samples derived from one biological sample. All statistical analyses were performed using GraphPad Prism eight software. Comparisons between two groups were performed by a two-tailed unpaired Student's *t*-test, Welch's *t*-test (in case *F*-test is significant: $p < 0.05$), or multiple *t*-test (for grouped data). Comparisons between multiple groups were performed by one or two-way analysis of variance with Tukey's multiple comparison. Survival of DSS-treated mice was analyzed by Kaplan–Meier survival plot using log-rank test. In all cases, *$p < 0.05$; **$p < 0.01$; ***$p < 0.001$; ****$p < 0.0001$; ns, not significant ($p > 0.05$).

## Acknowledgements

We thank Drs S Shioda, Y Niino, T Nakamachi, and J Watanabe (Showa University) for *Prkcd*$^{fl/fl}$ mice; K Nakayama (Kyushu University) for *Prkcd*$^{-/-}$ mice; K Rajewsky (Max Delbrück Center for Molecular Medicine) for *B1-8*$^{flox/+}$ mice and *Cd19*$^{cre/+}$ mice; M Nussenzweig (Rockefeller University) for *B1-8*$^{hi}$ mice; F Alt (Harvard Medical School) and T Tsubata (Tokyo Medical and Dental University) for *Igk*$^{-/-}$ mice; and P Burrows for critical reading of the manuscript.

## Additional information

### Funding

| Funder | Grant reference number | Author |
| --- | --- | --- |
| Japan Society for the Promotion of Science | Grant-in-Aid for Early-Career Scientists 19K16700 | Saori Fukao |

The funders had no role in study design, data collection and interpretation, or the decision to submit the work for publication.

### Author contributions

Saori Fukao, Conceptualization, Data curation, Formal analysis, Funding acquisition, Investigation, Validation, Visualization, Writing - original draft; Kei Haniuda, Formal analysis, Investigation, Writing - review and editing; Hiromasa Tamaki, Investigation; Daisuke Kitamura, Supervision, Writing - review and editing

### Author ORCIDs

Daisuke Kitamura (iD) http://orcid.org/0000-0002-5195-0474

### Ethics

All mice were maintained in the Tokyo University of Science (TUS) mouse facility under specific pathogen-free conditions. Mouse procedures were performed under protocols approved by the TUS Animal Care and Use Committee (Approval No.: S19017, S20011).

### Decision letter and Author response

Decision letter https://doi.org/10.7554/eLife.72116.sa1
Author response https://doi.org/10.7554/eLife.72116.sa2

## Additional files

### Supplementary files
• Supplementary file 1. The list of cytokines and primers used in this study.

• Transparent reporting form

### Data availability
All data generated or analysed during this study are included in the manuscript and supporting file; Source Data files have been provided for Figures 1-6, and figure supplements for Figures 1-3 and 5.

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
