## [Editor Report]

Although how structurally different antigens can activate B-lymphocytes to produce IgG with or without T cell help has been a long-standing question, this manuscript provides a clue to this important issue.

---

## [Decision Letter]

**Decision letter after peer review:**

Thank you for submitting your article "Protein kinase Cδ is essential for the IgG response against T cell-independent type 2 antigens and commensal bacteria" for consideration by *eLife*. Your article has been reviewed by 2 peer reviewers, and the evaluation has been overseen by a Reviewing Editor and Carla Rothlin as the Senior Editor. The following individuals involved in review of your submission have agreed to reveal their identity: Tomoharu Yasuda (Reviewer #1); Jurgen Wienands (Reviewer #2).

Essential revisions:

1) As PKCδ is known as a key signaling molecule suppressing autoimmune diseases. Important role of PKCδ in TI-2 response may suggest that homeostatic TI-2 response possibly influence on suppression of autoimmune diseases. Though authors briefly mentioned about autoimmunity in p 24, line 430-434, discussion may not be enough. For example, Michel Nussenzweig's group reported that critical role of Lyn in the regulation of clonal expansion and terminal differentiation of B cells in TI-2 responses (Shih et al., Role of antigen receptor affinity in T cell-independent antibody responses in vivo., Nat. Immunol., 3, 399, 2002). Since Lyn-deficient mice show autoimmune diseases similar to PKCδ-deficient mice (Nishizumi et al., Impaired proliferation of peripheral B cells and indication of autoimmune disease in lyn-deficient mice., Immunity, 3, 549, 1995), both Lyn and PKCδ may regulate same signaling pathway. Including further discussion related between TI-2 antigen/BCR signal and autoimmunity will enhance discussion.

2) Why and how is PKCδ required differently downstream of TI-2 BCR crosslinking but not in TD antigen exposure? It would be nice if authors discuss about this point.

3) How do IL-1α/β and IFNα regulate Aicda gene expression in synergy with BATF? It would be nice if authors discuss about transcriptional regulation of Aicda based on past reports.

4) How do TI-2 antigens engage PKC-δ? Is it solely the configuration or clustering of the BCR (as hypothesized in the Discussion part) or do secondary signals add to this and to the BATF-AID axis as suggested by the results of the forced expression of BATF to substitute for the loss of PKC-δ?

*Reviewer #1 (Recommendations for the authors):*

1) PKCδ is known as a key signaling molecule suppressing autoimmune diseases. Important role of PKCδ in TI-2 response may suggest that homeostatic TI-2 response possibly influence on suppression of autoimmune diseases. Though authors briefly mentioned about autoimmunity in p 24, line 430-434, discussion may not be enough. For example, Michel Nussenzweig's group reported that critical role of Lyn in the regulation of clonal expansion and terminal differentiation of B cells in TI-2 responses (Shih et al., Role of antigen receptor affinity in T cell-independent antibody responses in vivo., Nat. Immunol., 3, 399, 2002). Since Lyn-deficient mice show autoimmune diseases similar to PKCδ-deficient mice (Nishizumi et al., Impaired proliferation of peripheral B cells and indication of autoimmune disease in lyn-deficient mice., Immunity, 3, 549, 1995), both Lyn and PKCδ may regulate same signaling pathway. Including further discussion related between TI-2 antigen/BCR signal and autoimmunity will enhance discussion.

2) Why and how is PKCδ required differently downstream of TI-2 BCR crosslinking but not in TD antigen exposure? It would be nice if authors discuss about this point.

3) How do IL-1α/β and IFNα regulate Aicda gene expression in synergy with BATF? It would be nice if authors discuss about transcriptional regulation of Aicda based on past reports.

4) Figure 1C, 2B; Concentration unit should be converted to ng/ml same with other figures for IgM and IgG.

5) Figure 1C-E; NP-Ficoll stimulation in the absence of cytokines will be helpful to know background levels otherwise it is difficult to know the impact of IL-1 and IFNα in class switch recombination and antibody secretion.

*Reviewer #2 (Recommendations for the authors):*

1) A key aspect of this manuscript is the specificity with which PKC-δ is involved in TI-2 antigen responses as opposed to TD-dependent responses. In fact, Figure 2A and others suggest that PKC-δ activation is not an all-or-nothing event. Hence, the specificity or kinetic of PKC-δ activation could be corroborated. One important set of data is shown in Supplementary Figure 3. Why is the IgM response so strongly increased in the absence of PKC-δ?

2) How do TI-2 antigens engage PKC-δ? Is it solely the configuration or clustering of the BCR (as hypothesized in the Discussion part) or do secondary signals add to this and to the BATF-AID axis as suggested by the results of the forced expression of BATF to substitute for the loss of PKC-δ? The arrow head downstream of cytokine and TOLL-like receptors ends diffusely somewhere at the BATF-Aicda axis (Figure 7).

3) In other publications that describe the generation and analysis of PKC-δ-deficient mouse mutants such differences were not observed or less obvious. Is there an explanation to these differences?

---

## [Author Response]

Essential revisions:1) As PKCδ is known as a key signaling molecule suppressing autoimmune diseases. Important role of PKCδ in TI-2 response may suggest that homeostatic TI-2 response possibly influence on suppression of autoimmune diseases. Though authors briefly mentioned about autoimmunity in p 24, line 430-434, discussion may not be enough. For example, Michel Nussenzweig's group reported that critical role of Lyn in the regulation of clonal expansion and terminal differentiation of B cells in TI-2 responses (Shih et al., Role of antigen receptor affinity in T cell-independent antibody responses in vivo., Nat. Immunol., 3, 399, 2002). Since Lyn-deficient mice show autoimmune diseases similar to PKCδ-deficient mice (Nishizumi et al., Impaired proliferation of peripheral B cells and indication of autoimmune disease in lyn-deficient mice., Immunity, 3, 549, 1995), both Lyn and PKCδ may regulate same signaling pathway. Including further discussion related between TI-2 antigen/BCR signal and autoimmunity will enhance discussion.

We assume that dysregulation of microbiota due to the defective TI-2 response may cause autoimmunity, in conjunction with the defect of B-cell tolerance in PKCδ-deficient mice, as previously suggested in another mouse model (Manfredo Vieira et al., 2018); the defect of homeostatic production of the anti-bacterial IgG3 (Figure 6) may allow tissue translocation of some gut bacteria, which may in turn trigger autoantibody production in PKCδ-deficient mice. We have now included further discussion about this possibility (page 16, line 362-375).

It was shown that PKCδ is dispensable for BCR-stimulated activation and proliferation of B cells but is necessary to make self-reactive B cells unresponsive to the BCR stimulation (Mecklenbräuker et al., 2002). It was also reported that PKCδ exerts pro-apoptotic activity in normal peripheral B cells through nuclear translocation, unless BAFF signaling reverses it (Mecklenbräuker et al., 2004). On the other hand, Src-family tyrosine kinases including Lyn are redundantly required for BCR signal transduction, such as phosphorylation of key signaling molecules such as Igα/β and CD19. In addition, Lyn is also responsible for phosphorylation and activation of inhibitory co-receptors for BCR such as CD22 and FcγRIIb, which is the basis for spontaneous activation and exaggerated responses of B cells leading to autoimmunity in Lyn-deficient mice. Therefore, the mechanism for the induction of autoimmunity in PKCδ-deficient mice appears quite different from that in Lyn-deficient mice. Also, IgG production in response to TI-2 Ag or bacteria has not been studied in Lyn-deficient mice. Thus, we have not referred to Lyn-deficient mice in this manuscript.

2) Why and how is PKCδ required differently downstream of TI-2 BCR crosslinking but not in TD antigen exposure? It would be nice if authors discuss about this point.

As previously appreciated, a role of BCR signaling is substantially different between TI-2 and TD responses: Tyrosine kinases Btk and an adaptor molecule BLNK, for example, are necessary for TI-2, but not TD, immune response (as we wrote in page 2, the second paragraph). Unlike TI-2 antigen, TD antigen alone cannot induce proliferation, IgM production, and Batf expression in naive B cells, as we demonstrated in *in-vitro* study (Figures 1A, 1B, 5B, Figure 5 —figure supplement 1A). Thus, it appears that TD antigen-mediated BCR signaling cannot exceed a threshold for B-cell activation including sufficient phosphorylation of PKCδ (Figure 2A) required for class switching to IgG; T-cell-derived signals are known to compensate for the defective BCR signaling in the TD response. We have now included this discussion in the Discussion section (page 14, line 313-321).

We still don’t know how PKCδ is activated and exert its function after BCR crosslinking by TI-2 antigen. The phosphorylation of Thr 505 and Ser 647 of PKCδ, known to be important for its enzymatic activity, was observed in unstimulated naïve B cells and unchanged by TI-2 antigen stimulation in our *in-vitro* system (data not shown), but that of Tyr 311 was augmented, which is known to alter its intracellular localization and the substrate specificity (Rybin et al., 2004; Steinberg, 2004). Thus, identification of the substrate of PKCδ in B cells after stimulation with TI-2 antigen would be an important next step. We have now included this discussion (page 13, line 290 – page 14, line 312).

3) How do IL-1α/β and IFNα regulate Aicda gene expression in synergy with BATF? It would be nice if authors discuss about transcriptional regulation of Aicda based on past reports.

It has been shown that the transcription of *Aicda* requires the activation of the 5’ upstream enhancer region of *Aicda* locus, in addition to the recruitment of BATF to the 3’ downstream region in the context of TD response (Tran et al., 2010; Vaidyanathan et al., 2014). Therefore, we consider that the secondary stimulation such as IL-1α/β and IFNα might signal to activate the 5’ upstream enhancer region and induce the transcription of *Aicda* in synergy with BATF induced by TI-2 antigen stimulation. We have now described about this hypothesis in detail in the Discussion section (page 14, line 325 – page 15, line 340).

4) How do TI-2 antigens engage PKC-δ? Is it solely the configuration or clustering of the BCR (as hypothesized in the Discussion part) or do secondary signals add to this and to the BATF-AID axis as suggested by the results of the forced expression of BATF to substitute for the loss of PKC-δ?

We have not investigated how TI-2 antigen-bound BCR engages PKCδ. It would be challenging to study the relation between the configuration or clustering of BCR and the engagement/activation of PKCδ, which would require a specialized technology such as TIRF microscopy. As we have shown in the revised manuscript, TI-2 antigen alone could induce PKCδ phosphorylation and *Batf* expression, but IL-1α/β or IFNα alone could not, nor they could augment the TI-2 antigen-induced these responses (Figure 5B, the newly added ‘Figure 5 —figure supplement 1B-D’). Therefore, we consider that TI-2 antigen activates the PKCδ-BATF axis solely through BCR. We suppose that this signaling axis and the secondary signals may merge on the *Aicda* locus to induce AID expression as described in our response to the Essential Revisions 3. We have mentioned about the new data in the Result section (page 10, line 204-206).

Reviewer #1 (Recommendations for the authors):1) As I commented in Public review, PKCδ is known as a key signaling molecule suppressing autoimmune diseases. Important role of PKCδ in TI-2 response may suggest that homeostatic TI-2 response possibly influence on suppression of autoimmune diseases. Though authors briefly mentioned about autoimmunity in p 24, line 430-434, discussion may not be enough. For example, Michel Nussenzweig's group reported that critical role of Lyn in the regulation of clonal expansion and terminal differentiation of B cells in TI-2 responses (Shih et al., Role of antigen receptor affinity in T cell-independent antibody responses in vivo., Nat. Immunol., 3, 399, 2002). Since Lyn-deficient mice show autoimmune diseases similar to PKCδ-deficient mice (Nishizumi et al., Impaired proliferation of peripheral B cells and indication of autoimmune disease in lyn-deficient mice., Immunity, 3, 549, 1995), both Lyn and PKCδ may regulate same signaling pathway. Including further discussion related between TI-2 antigen/BCR signal and autoimmunity will enhance discussion.

This is the same comment as in the Essential Revisions 1, to which we have answered.

2) Why and how is PKCδ required differently downstream of TI-2 BCR crosslinking but not in TD antigen exposure? It would be nice if authors discuss about this point.

This is the same comment as in the Essential Revisions 2, to which we have answered.

3) How do IL-1α/β and IFNα regulate Aicda gene expression in synergy with BATF? It would be nice if authors discuss about transcriptional regulation of Aicda based on past reports.

This is the same comment as in the Essential Revisions 3, to which we have answered.

4) Figure 1C, 2B; Concentration unit should be converted to ng/ml same with other figures for IgM and IgG.

We have revised the Figure 2B to use ng/ml for IgM concentration. To measure IgG3 titers in the experiments shown in Figure 1C and other figures, we used serum of a C57BL/6 mouse as a standard, and therefore the values were shown as relative units (AU). It would not mean anyway if we had measured absolute values of the IgG3 concentrations; the values cannot be compared among the figures, since the number of cultured cells and the culture period varied among the figures, depending on the situation of each experiment such as the number of available mice.

5) Figure 1C-E; NP-Ficoll stimulation in the absence of cytokines will be helpful to know background levels otherwise it is difficult to know the impact of IL-1 and IFNα in class switch recombination and antibody secretion.

We have added back the data of stimulation with antigen alone in the revised Figure 1C-E. Consequently, the data previously shown as Figure 1 —figure supplement 1F has been removed since it has been integrated into the new Figure 1E.

Reviewer #2 (Recommendations for the authors):1) A key aspect of this manuscript is the specificity with which PKC-δ is involved in TI-2 antigen responses as opposed to TD-dependent responses. In fact, Figure 2A and others suggest that PKC-δ activation is not an all-or-nothing event. Hence, the specificity or kinetic of PKC-δ activation could be corroborated. One important set of data is shown in Supplementary Figure 3. Why is the IgM response so strongly increased in the absence of PKC-δ?

As this comment is essentially integrated in the Essential Revisions 2, please look at our response written there. In particular, we admit that TD-antigen stimulation modestly and transiently increased the PKCδ phosphorylation at Tyr 311 (Figure 2A) but did not induce Batf expression at all (Figure 5B, Figure 5 —figure supplement 1A). Therefore, the extent of PKCδ phosphorylation might not be enough to exceed the activation threshold.

We think the increased IgM response in the TD immune response of PKCδ-deficient mice reflects the reported constitutive pro-apoptotic activity of PKCδ in the peripheral B cells (Mecklenbräuker et al., 2004) that may normally limit the initial proliferation of B cells in the TD immune response. We have mentioned this possibility in the Discussion section (page 13, line 294 – page 14, 307).

2) How do TI-2 antigens engage PKC-δ? Is it solely the configuration or clustering of the BCR (as hypothesized in the Discussion part) or do secondary signals add to this and to the BATF-AID axis as suggested by the results of the forced expression of BATF to substitute for the loss of PKC-δ? The arrow head downstream of cytokine and TOLL-like receptors ends diffusely somewhere at the BATF-Aicda axis (Figure 7).

This is the same comment as in the Essential Revisions 4, to which we have answered.

3) In other publications that describe the generation and analysis of PKC-δ-deficient mouse mutants such differences were not observed or less obvious. Is there an explanation to these differences?

Although we have carefully searched for the literature, we could not find any studies analyzing immune responses or class-switching in PKCδ-deficient mice. Most of the studies were focused on B cell tolerance.